# A Differentially Private Linear-Time fPTAS for the Minimum Enclosing Ball Problem

**Bar Mahpud**      **Or Sheffet**
Faculty of Engineering
Bar-Ilan University, Israel
{mahpudb, or.sheffet}@biu.ac.il

## Abstract

The Minimum Enclosing Ball (MEB) problem is one of the most fundamental problems in clustering, with applications in operations research, statistics and computational geometry. In this works, we give the first linear time differentially private (DP) fPTAS for the Minimum Enclosing Ball problem, improving both on the runtime and the utility bound of the best known DP-PTAS for the problem, of Ghazi et al [21]. Given $n$ points in $\mathbb{R}^d$ that are covered by the ball $B(\theta_{opt}, r_{opt})$, our simple iterative DP-algorithm returns a ball $B(\theta, r)$ where $r \leq (1 + \gamma)r_{opt}$ and which leaves at most $\tilde{O}(\frac{\sqrt{d}}{\gamma\epsilon})$ points uncovered in $\tilde{O}(n/\gamma^2)$-time. We also give a local-model version of our algorithm, that leaves at most $\tilde{O}(\frac{\sqrt{nd}}{\gamma\epsilon})$ points uncovered, improving on the $n^{0.67}$-bound of Nissim and Stemmer [31] (at the expense of other parameters). Lastly, we test our algorithm empirically and discuss open problems.

## 1   Introduction and Related Work

One of the fundamental problems in clustering is the Minimum Enclosing Ball (MEB) problem, or the 1-Center problem, in which we are given a dataset $P \subset \mathbb{R}^d$ containing $n$ points, and our goal is to find the smallest possible ball $B(\theta_{opt}, r_{opt})$ that contains $P$. The MEB problem has applications in various areas of operations research, machine learning, statistics and computational geometry: gap tolerant classifiers [11], tuning Support Vector Machine parameters [12] and Support Vector Clustering [5, 6], $k$-center clustering [10], solving the approximate 1-cylinder problem [10], computation of spatial hierarchies (e.g., sphere trees [25]), and others [18]. The MEB problem is NP-hard to solve exactly, but it can be solved in linear time in constant dimension [29, 19] and has several fully-Polynomial Time Approximation Schemes (fPTAS) [4, 27] that approximate it to any constant $(1 + \gamma)$ in time $O(n/\gamma)$; as well as an additive $\gamma$ approximation in sublinear time [13].

But in situations where the data is sensitive in nature, such as addresses, locations or descriptive feature-vectors[1] we run the risk that approximating the data's MEB might leak information about a single individual. Differential privacy [16, 15] (DP) alleviates such a concern as it requires that no single individual has a significant effect on the output. Alas, the MEB problem is highly sensitive in nature, since there exist datasets where a change to a single datum may affect the MEB significantly.

In contrast, it is evident that for any fixed ball $B(\theta, r)$ the number of input points that $B$ contains changes by no more than one when changing any single datum. And so, in DP we give *bi-criteria* approximations of the MEB: a ball $B(\theta, r)$ that may leave at most a few points of $P$ uncovered and whose radius is comparable to $r_{opt}$. The work of [32] returns a $O(\sqrt{\log(n)})$-approximation

---

[1]Consider a research in a hospital in which one first runs some regression on each patient's data, and then looks for the spread of all regressors of all patients.

36th Conference on Neural Information Processing Systems (NeurIPS 2022).

of the MEB while omitting as few as $\tilde{O}(1/\epsilon)$ points from $P$, and it was later improved to a $O(1)$-approximation [31]. The work of [21] does give a PTAS for the MEB problem, but their $(1+\gamma)$-approximation may leave $\tilde{O}(\sqrt{d}/\epsilon\gamma^3)$ datapoints uncovered[2] and it runs in $n^{O(1/\gamma^2)}$-time where the constant hidden in the big-$O$ notation is huge; as it leverages on multiple tools that take $\exp(\dim)$-time to construct, such as almost-perfect lattices and list-decodable covers. It should be noted that all of these works actually study the related problem of 1-cluster in which one is given an additional parameter $t$ and seeks to find the smallest MEB of a subset $Q \subset P$ where $|Q| \geq t$. Lastly (as was first commented in [21], Section D.2.1.), a natural way to approximate the MEB problem is through minimizing the convex hinge-loss $L(\theta, x) = \frac{1}{r}\max\{0, \|x-\theta\| - r\}$ but its utility depends on $r$ (as the utility of DP-ERM scales with the Lipfshitz constant of the loss [3]).[3]

By far, one of the most prominent uses of the DP-approximations of the MEB problem lies in range estimation, as $O(1)$-approximations of the MEB can assist in reducing an a-priori large domain to a ball whose radius is proportional to the diameter of $P$. This helps in reducing the $L_2$-sensitivity of problems such as the mean and other distance related queries (e.g. PCA). So for example, if we have $\tilde{\Omega}(\frac{\sqrt{d}}{\gamma\epsilon})$ points in a ball of radius $10r_{opt}$ then a DP-approximation of the data's mean using the Gaussian mechanism (see Section 2) returns a point of distance $\leq \gamma r_{opt}$ to the true mean (a technique that is often applied in a Subsample-and-Aggregate framework [30]). This averaging also gives an efficient $(2+\gamma)$-approximation of the MEB. But it is still unknown whether there exists a DP $c$-approximation of the MEB for $c < 2$ whose runtime is below, say, $n^{100}$.

**Our Contribution and Organization.** In this work, we give the first DP-fPTAS for the MEB problem. Our algorithm is very simple and so is its analysis. As input, we assume the algorithm is run after the algorithms of [31] were already run, and as a "starting point" we have both (a) a real number $r_0$ which is a $4$-approximation of $r_{opt}$, and (b) a $10$-approximation of the MEB itself, namely a ball $B$ such that $P \subset B$,[4] which is centered at a point $\theta_0$ satisfying $\|\theta_0 - \theta_{opt}\| \leq 10r_{opt}$.[5] It is now our goal to refine these parameters to a $(1+\gamma)$-approximation of the MEB. In fact, we can assume that we have a $(1+\gamma)$-approximation of the value of $r_{opt}$: we simply iterate over all powers: $\frac{r_0}{4}, \frac{r_0}{4}(1+\gamma), \frac{r_0}{4}(1+\gamma)^2, ..., r_0$ where for each guess of $r$ we apply a privacy preserving procedure returning either a point $\theta$ satisfying $P \subset B(\theta, r)$ or $\perp$. In our algorithm we simply use a binary-search over these $O(1/\gamma)$ possible values, in order to save on the privacy-budget.

Now, given $\theta_0$ and some radius-guess $r$, our goal is to shift $\theta_0$ towards $\theta_{opt}$. So, starting from $\theta^0 = \theta_0$, we repeat this simple iterative procedure: we take the mean $\mu$ of the points *uncovered by the current* $B(\theta^t, r)$ and update $\theta^{t+1} \leftarrow \theta^t + \frac{\gamma^2}{2}(\mu - \theta^t)$. We argue that, if $r \geq r_{opt}$ then after $T = O(\gamma^{-2}\log(1/\gamma))$-iterations we get $\theta^T$ such that $\|\theta^T - \theta_{opt}\| \leq \gamma r_{opt}$ and therefore have that $P \subset B(\theta^T, (1+\gamma)r)$. The reason can be easily seen from Figure 1 — any point $x \in P$ which is uncovered by the current $B(\theta^t, r)$ must be closer to $\theta_{opt}$ than to $\theta^t$, and therefore must have a noticeable projection onto the direction $\theta_{opt} - \theta^t$. Thus, in a Perceptron-like style, making a $\Theta(\gamma^2)$-size step towards this $x$ must push us significantly in the $\theta_{opt} - \theta^t$ direction. We thus prove that if the distance of $\theta^t$ from $\theta_{opt}$ is large, this update step reduces our distance to $\theta_{opt}$. Note that our proof shows that in the non-private case it suffices to take any uncovered point in order to make this progress, or any convex-combination of the uncovered points.

In the private case, rather than using the true mean of the uncovered points in each iteration, we have to use an approximated mean. So we prove that applying our iterative algorithm with a "nice" distribution whose mean has a large projection in the $\theta_{opt} - \theta^t$ direction also returns a good $\theta^T$ in expectation, and then amplify the success probability by naïve repetitions. We also give a SQ-style algorithm for approximating the MEB under proximity conditions between the true- and the noisy-mean, a result which may be of interest by itself. After discussing preliminaries in Section 2, we present both the standard (non-noisy) version of our algorithm and its noisy variation in Section 3.

Having established that our algorithm works even with a "nice" distribution whose mean approximates the mean of the uncovered points, all that is left is just to set the parameters of a privacy preserving

---

[2]See Lemmas 59 & 60 in [21]

[3]In fact, there's more to this discussion, as we detail at the end of the introduction.

[4]We can always omit the few input points that may reside outside this ball.

[5]We comment that replacing these $4$ and $10$ constants with any other constants merely changes the constants in our analysis in a very straight-forward way.

algorithm accordingly. To that end we work with the notion of zCDP [8] and apply solely the Gaussian mechanism. To obtain these nice properties, it follows that the number of uncovered points must be $\Omega(\sqrt{d}/\epsilon^t)$ where $\epsilon^t$ is the privacy budget of the $t^{\text{th}}$-iteration, or else we halt. And due to the composition theorem of DP it suffices to set $\epsilon^t = O(\epsilon/\sqrt{T})$. This leads to a win-win situation: either we find in some iteration a ball that leaves no more than $\tilde{O}(\sqrt{d}/\gamma\epsilon)$ points uncovered, or we complete all iterations and obtain a ball of radius $\leq (1+\gamma)r$ that covers all of $P$. The full details of this analysis appear in Section 4. We then repeat this analysis but in the local-model, where each user adds Gaussian noise to her own input point. This leads to a similar analysis incurring a $\sqrt{n}$-larger bounds, as detailed in Section 5.

While at the topic of local-model DP (LDP) algorithms, it is worth mentioning that the algorithms of [31], which provide us with a good initial "starting point", do have a LDP-variant. Yet the LDP variants of these algorithms may leave as many as $n^{0.67}$ datapoints uncovered. So in Appendix A we give simple differentially private algorithms (in both the curator- and local-models) that obtain such good $\theta_0$ and $r_0$. Formally, our LDP-algorithm returns a ball $B(\theta_0, r_0)$ s.t. by projecting all points in $P$ onto $B(\theta_0, r)$ we alter no more than $\tilde{O}(\sqrt{d}/\epsilon)$ points and obtain $P' \subset B(\theta_0, r_0)$ where $r_0 \leq 6r_{opt}(P')$. Thus, combining our LDP algorithm for finding a good starting point together with the algorithm of Section 5 we get an overall $(1+\gamma)$-approximation of the MEB in the local model which may omit / alter as many as $\tilde{O}(\sqrt{nd}/\gamma\epsilon)$-points. We comment that while this improves on the previously best-known LDP algorithm's bound of $n^{0.67}$, our algorithm's dependency on parameters such as the dimension $d$ or grid-size[6] is worse, and furthermore – that the analysis of [31] (i) relates to the problem of 1-cluster (finding a cluster containing $t \leq n$ many points) and (ii) separates between the required cluster size and the number of omitted points (which is much smaller and only logarithmic in $d$), two aspects that are not covered in our work.

Lastly, we provide empirical evaluations of our algorithm (which we deferred in its entirety to the Supplementary Material, Section E) showing a rather ubiquitous performance across multiple datasets, and discuss open problems in Section 6.

**Comparison with the ERM Baseline.** Recall that the MEB problem, given a suggested radius $r$ and a convex set $\Theta$, can be formulated as a ERM problem using a hinge-loss function $\ell^1(\theta; x) = \max\{0, \frac{\|x-\theta\|-r}{\text{diam}(\Theta)}\}$. Indeed, when $\text{diam}(\Theta) \gg r$ then privately solving this ERM problem gives no useful guarantee about the result, but much like our algorithm one can first find some $\theta_0$ close up to, say, $10r$ to $\theta_{opt}$ and set $\Theta$ as a ball of radius $O(r)$. Since there exists $\theta_{opt}$ for which $\frac{1}{n}\sum_x \ell^1(\theta; x) = 0$, then private SGD [3, 2] returns $\tilde{\theta}$ for which $\frac{1}{n}\sum_x \ell^1(\tilde{\theta}; x) \leq C\frac{\sqrt{d}}{\epsilon n}$ for some constant $C > 0$. This upper-bounds the number of points that contribute $\gamma r$ to this loss at $C\frac{\sqrt{d}}{\epsilon\gamma}$, and so $|P \setminus B(\tilde{\theta}, (1+\gamma)r)| = O(\frac{\sqrt{d}}{\epsilon\gamma})$. However, the caveat is that the SGD algorithm achieves such low loss using $O(n^2)$-SGD iterations.[7] In contrast our analysis can be viewed as proving that for the equivalent ERM in the square of the norm, $\ell^2(\theta; x) = \max\{0, \frac{\|x-\theta\|^2-r^2}{\text{diam}(\Theta)^2}\}$, it suffices to make only $\tilde{O}(\gamma^{-2})$ *non-zero* gradient steps to have some $\theta^T$ s.t. $\|\theta^T - \theta_{opt}\| \leq \gamma r$ so that $B(\theta^T, (1+\gamma)r)$ covers all of the input. Thus, our result is obtained in linear $\tilde{O}(n/\gamma^2)$-time.

## 2 Preliminaries

**Notation.** Given a vector $v \in \mathbb{R}^d$ we denote its $L_2$-norm as $\|v\|$, and also use $\langle v, u \rangle$ to denote the dot-product between two $d$-dimensional vectors $u$ and $v$. A (closed) ball $B(\theta, r)$ is the set of all points $B(\theta, r) = \{x \in \mathbb{R}^d : \|x - \theta\| \leq r\}$. We use $\tilde{O}(\cdot)$ / $\tilde{\Omega}(\cdot)$ to denote big-$O$ / big-$\Omega$ dependency up to $\text{poly}\log$ factors. We comment that in our work we made no effort to optimize constants.

**The Gaussian and $\chi_d^2$-Distributions.** Given two parameters $\mu \in \mathbb{R}$ and $\sigma^2 > 0$ we denote $\mathcal{N}(\mu, \sigma^2)$ as the Gaussian distribution whose PDF at a point $x \in \mathbb{R}$ is $(2\pi\sigma^2)^{0.5}\exp(-\frac{(x-\mu)^2}{2\sigma^2})$. Standard concentration bounds give that for any $x > 1$ the probability $\Pr_{X \sim \mathcal{N}(\mu, \sigma^2)}[|X - \mu| \geq x\sigma] \leq 2\exp(-x^2/2)$. It is well-known that given two independent random variable $X \sim \mathcal{N}(\mu_1, \sigma_1^2)$

---

[6]It is known [7] that DP MEB-approximation requires the input points to lie on some prespecified finite grid.
[7]Unfortunately, the hinge-loss isn't smooth, ruling out the linear SGD of [20].

and $Y \sim \mathcal{N}(\mu_2, \sigma_2^2)$ their sum is distributed like a Gaussian $X + Y \sim \mathcal{N}(\mu_1 + \mu_2, \sigma_1^2 + \sigma_2^2)$. We also denote $\mathcal{N}(v, \sigma^2 I_d)$ as the distribution over $d$-dimensional vectors where each coordinate $j$ is drawn i.i.d. from $\mathcal{N}(v_j, \sigma^2)$. Given $X \sim \mathcal{N}(0, \sigma^2 I_d)$ it is known that $\|X\|^2$ is distributed like a $\chi_d^2$-distribution; and known concentration bounds on the $\chi_d^2$-distribution give that for any $x > 1$ the probability $\Pr_{X \sim \mathcal{N}(0, \sigma^2 I_d)}[\|X\|^2 > \sigma^2(\sqrt{d} + x)^2] \leq \exp(-x^2/2)$.

**Differential Privacy.** Given a domain $\mathcal{X}$, two multi-sets $P, P' \in \mathcal{X}^n$ are called *neighbors* if they differ on a single entry. An algorithm (alternatively, mechanism) $\mathcal{M}$ is said to be $(\epsilon, \delta)$-*differentially private* (DP) [16, 15] if for any two neighboring $P, P'$ and any set $S$ of possible outputs we have: $\Pr[\mathcal{M}(P) \in S] \leq e^\epsilon \Pr[\mathcal{M}(P') \in S] + \delta$.

An algorithm is said to be $\rho$-zero concentrated differentially privacy (zCDP) [8] if for and two neighboring $P$ and $P'$ and any $\alpha > 1$, the $\alpha$-Réyni divergence between the output distribution of $\mathcal{M}(P)$ and of $\mathcal{M}(P')$ is upper bounded by $\alpha\rho$, namely

$$\forall \alpha > 1, \ \frac{1}{\alpha - 1} \log \left( \underset{x \sim \mathcal{M}(P')}{\mathbb{E}} \left[ \left( \frac{\mathsf{PDF}[\mathcal{M}(P) = x]}{\mathsf{PDF}[\mathcal{M}(P') = x]} \right)^\alpha \right] \right) \leq \alpha\rho$$

It is a well-known fact that the composition of two $\rho$-zCDP mechanisms is $2\rho$-zCDP. It is also known that given a function $f : \mathcal{X}^n \to \mathbb{R}^d$ whose $L_2$-global sensitivity is $\max_{P \sim P'} \|f(P) - f(P')\|_2 \leq G$ then the Gaussian mechanism that returns $f(D) + X$ where $X \sim \mathcal{N}(0, \frac{G^2}{2\rho} I_d)$ is $\rho$-zCDP. Lastly, it is known that any $\rho$-zCDP mechanism is $(\epsilon, \delta)$-DP for any $\delta < 1$ and $\epsilon = \rho + \sqrt{4\rho \ln(1/\delta)}$. This suggests that given $\epsilon \leq 1$ and $\delta \leq e^{-2}$ it suffices to use a $\rho$-zCDP mechanis with $\rho \leq \frac{\epsilon^2}{5 \ln(1/\delta)}$.

The *Local-Model* of DP: while standard algorithms in DP assume the existence of a trusted curator who has access to the raw data, in the local-model of DP no such curator exists. While the formal definition of the local-model involves the notion of protocols (see [35] for a formal definition), for the context of this work it suffices to say each respondent randomized her own messages so that altogether they preserve $\rho$-zCDP.

## 3 A Non-Private fPTAS for the MEB Problem

In this section we give our non-private algorithm. We first analyze it assuming no noise – namely, in each iteration we use the precise mean of the points that do not reside inside the ball $B(\theta^t, r)$. Later, in Section 3.1 we discuss a version of this algorithm in which rather than getting the exact mean, we get a point which is sufficiently close to the mean.

---

**Algorithm 1** Non-Private Minimum Enclosing Ball

---

**Input:** a set of $n$ points $P \subseteq \mathbb{R}^d$, an approximation parameter $\gamma \in (0, 1)$,
an initial radius $r_0$ s.t. $r_{opt} \leq r_0 \leq 4r_{opt}$, and an initial center $\theta_0$ s.t. $\|\theta_0 - \theta_{opt}\| \leq 10r_{opt}$.

1: Set $i_{\min} \leftarrow 0$, $i_{\max} \leftarrow \ln_{1+\gamma}(4)(\approx \frac{4}{\gamma})$, and $\theta^* \leftarrow \theta_0$.
2: **while** $(i_{\min} < i_{\max})$ **do**
3:      $i_{cur} = \lfloor \frac{i_{\min} + i_{\max}}{2} \rfloor$
4:      $r_{cur} \leftarrow (1 + \gamma)^{i_{cur}} \cdot r_0/4$
5:      $\theta_{cur} \leftarrow \text{MMEB}(P, \gamma, r_{cur}, \theta_0)$
6:      **if** $P \subset B(\theta_{cut}, (1 + \gamma)r_{cur})$ **then**
7:          Set $i_{\max} \leftarrow i_{cur}$, $\theta^* \leftarrow \theta_{cur}$ and $r^* \leftarrow (1 + \gamma)r_{cur}$
8:      **else**
9:          $i_{\min} \leftarrow i_{cur} + 1$
10: **return** $B(\theta^*, r^*)$

---

**Theorem 3.1.** *For any $P \subset \mathbb{R}^d$, denote $B(\theta_{opt}, r_{opt})$ as the MEB of $P$. Then Algorithm 1 returns a ball $B(\theta, r)$ where $P \subset B(\theta, r)$ and $r \leq (1 + 3\gamma)r_{opt}$.*

At the core of the proof of Theorem 3.1 lies the following lemma.

**Lemma 3.2.** Applying Algorithm 2 with any $r \geq r_{opt}$ and any $\theta_0$ where $\|\theta_0 - \theta_{opt}\| \leq 10r_{opt}$ we obtain a $\theta$ where $\|\theta - \theta_{opt}\| \leq \gamma r_{opt}$ in at most $T$ iterations.

---
**Algorithm 2** Margin based Minimum Enclosing Ball (MMEB)
---

**Input:** a set of $n$ points $P \subseteq \mathbb{R}^d$, an approximation parameter $\gamma \in (0,1)$, a candidate radius $r$, and an initial center $\theta_0$ s.t. $\|\theta_0 - \theta_{opt}\| \le 10 r_{opt}$.

1: Set $T \leftarrow \frac{4}{\gamma^2} \ln(\frac{100}{\gamma^2})$, and $\theta^0 = \theta_0$.
2: **for** $t = 0, 1, 2, \ldots, T-1$ **do**
3:     **if** $(\{x \in P : x \notin B(\theta^t, r)\} = \emptyset)$ **then return** $\theta^t$
4:     **else**
5:         Set $n_w^t \leftarrow |\{x \in P : x \notin B(\theta^t, r)\}|$ and $\mu_w^t \leftarrow \frac{1}{n_w^t} \sum\limits_{x \notin B(\theta^t, r)} x$
6:         Update $\theta^{t+1} \leftarrow \theta^t - \frac{\gamma^2}{2}(\theta^t - \mu_w^t)$
7: **return** $\theta^T$

---

It is important to note that Lemma 3.2 holds even if in each iteration the update step isn't based on the mean $\mu^t$ of the set of uncovered point, but rather *any* convex combination of the uncovered points. Specifically, even if we use in each iteration a single point which is uncovered by $B(\theta^t, r)$, then the algorithm's convergence in $T$ steps can be guaranteed.

*Proof of Theorem 3.1.* Suppose Lemma 3.2 indeed holds. Then it immediately implies whenever Algorithm 2 is run with $r \ge r_{opt}$ we obtain a point $\theta$ where $P \subset B(\theta_{opt}, r_{opt}) \subset B(\theta, (1+\gamma) r_{opt})$. Denote $i^* = \min\{i \in \mathbb{N} : \frac{r_0}{4}(1+\gamma)^i \ge r_{opt}\}$. It is simple to prove inductively that in each iteration of Algorithm 1 we have that $i^* \ge i_{\min}$. Next, call an integer $i$ successful if we obtain for its radius $r_{cur}(i)$ some point $\theta$ where $P \subset B(\theta, (1+\gamma) r_{cur}(i))$. Again, it is simple to argue inductively that $i_{\max}$ is always successful. It follows that when the binary search of Algorithm 1 terminates, $i_{\min} = i_{\max}$ and we have a successful $i$, and so we return a ball of radius $\frac{r_0}{4}(1+\gamma)^{i_{\min}} \cdot (1+\gamma) \le (1+\gamma)^2 r_{opt} \le (1+3\gamma) r_{opt}$ which contains all points in $P$, thus concluding our proof. $\square$

Thus, all that is left is to prove Lemma 3.2. Its proof, in turn, requires the following claim.

**Claim 3.3.** Given a set of $n$ points $P \subseteq \mathbb{R}^d$, let $B(\theta_{opt}, r_{opt})$ denote the MEB of $P$. Let $\theta \in \mathbb{R}^d$ be an arbitrary point, and let $r$ be any real number where $r \ge r_{opt}$. Then for any $x \in P$ s.t. $\|\theta - x\| > r$ it holds that

$$\langle \theta - \theta_{opt}, x - \theta_{opt} \rangle \le \frac{1}{2} \|\theta - \theta_{opt}\|^2$$

*Proof.* Let $x \in P$ be a point s.t. $x \notin B(\theta, r)$, as depicted in Figure 1. Let $m$ be the middle point $\frac{\theta + \theta_{opt}}{2}$, and let $\mathcal{H}$ be the hyperplane orthogonal to $\theta - \theta_{opt}$ which passes through $m$. Denote $\mathcal{H}^+$ as the (open) halfspace $\mathcal{H}^+ = \{z \in \mathbb{R}^d : \|z - \theta_{opt}\| < \|z - \theta\|\}$. Therefore $x \in \mathcal{H}^+$ which in turn implies that

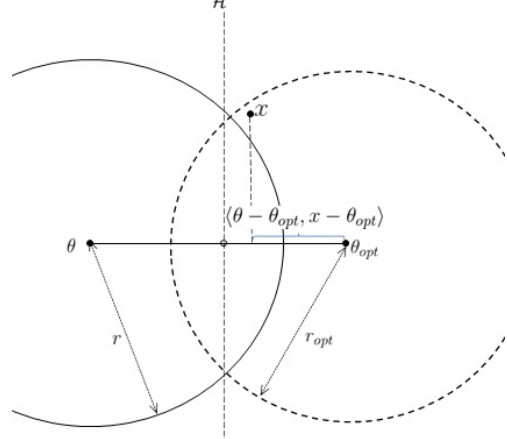

Figure 1: For a point $x$ uncovered by $B(\theta, r)$ where $r \ge r_{opt}$, it must be that $x$'s projection onto the $\overline{\theta \theta_{opt}}$-line is closer to $\theta_{opt}$ than to $\theta$.

$$\langle x - \theta_{opt}, \theta - \theta_{opt} \rangle < \langle m - \theta_{opt}, \theta - \theta_{opt} \rangle = \frac{1}{2} \|\theta - \theta_{opt}\|^2 \qquad \square$$

We are now ready to prove our main lemma.

*Proof of Lemma 3.2.* First, we argue that in any iteration $t$ of Algorithm 2 where $\{x \in P : x \notin B(\theta^t, r)\} \ne \emptyset$ it holds that $\|\theta^{t+1} - \theta_{opt}\|^2 \le (1 - \frac{\gamma^2}{2})\|\theta^t - \theta_{opt}\|^2 + (\frac{\gamma^2}{2})^2 \cdot r_{opt}^2$. That is because by definition

$$\|\theta^{t+1} - \theta_{opt}\|^2 = \left\| \left( (1 - \frac{\gamma^2}{2})\theta^t + \frac{\gamma^2}{2}\mu_w^t \right) - \theta_{opt} \right\|^2 = \left\| (1 - \frac{\gamma^2}{2})(\theta^t - \theta_{opt}) + \frac{\gamma^2}{2}(\mu_w^t - \theta_{opt}) \right\|^2$$

$$= (1 - \frac{\gamma^2}{2})^2 \cdot \|\theta^t - \theta_{opt}\|^2 + 2\frac{\gamma^2}{2}(1 - \frac{\gamma^2}{2})\langle\theta^t - \theta_{opt}, \mu_{w^t} - \theta_{opt}\rangle + (\frac{\gamma^2}{2})^2 \cdot \|\mu_{w^t} - \theta_{opt}\|^2$$

Claim 3.3 gives that $\langle\theta^t - \theta_{opt}, \mu_w^t - \theta_{opt}\rangle = \frac{1}{n_w^t} \sum_{x \notin B(\theta^t, r)} \langle\theta^t - \theta_{opt}, x - \theta_{opt}\rangle \leq \frac{1}{2}\|\theta^t - \theta_{opt}\|^2$, so

$$\leq (1 - \frac{\gamma^2}{2})^2 \cdot \|\theta^t - \theta_{opt}\|^2 + 2(\frac{\gamma^2}{2} - \frac{\gamma^4}{4}) \cdot \frac{1}{2}\|\theta^t - \theta_{opt}\|^2 + (\frac{\gamma^2}{2})^2 \cdot \|\mu_{w^t} - \theta_{opt}\|^2$$

Lastly note that the ball $B(\theta_{opt}, r_{opt})$ is convex and so

$$\leq (1 - \gamma^2 + \frac{\gamma^4}{4}) \cdot \|\theta^t - \theta_{opt}\|^2 + (\frac{\gamma^2}{2} - \frac{\gamma^4}{4}) \cdot \|\theta^t - \theta_{opt}\|^2 + \frac{\gamma^4}{4} \cdot r_{opt}^2$$

$$\leq (1 - \frac{\gamma^2}{2})\|\theta^t - \theta_{opt}\|^2 + \frac{\gamma^4}{4} \cdot r_{opt}^2$$

$$(1)$$

So now, consider any iteration of Algorithm 2 with $r \geq r_{opt}$ and where $\|\theta^t - \theta_{opt}\| \geq \gamma r_{opt}$ and in which we make an update step. Due to Equation (1)

$$\|\theta^{t+1} - \theta_{opt}\|^2 \leq (1 - \frac{\gamma^2}{2})\|\theta^t - \theta_{opt}\|^2 + \frac{\gamma^4}{4} \cdot r_{opt}^2 \leq (1 - \frac{\gamma^2}{2})\|\theta^t - \theta_{opt}\|^2 + \frac{\gamma^4}{4} \cdot \frac{\|\theta^t - \theta_{opt}\|^2}{\gamma^2}$$

$$= (1 - \frac{\gamma^2}{4})\|\theta^t - \theta_{opt}\|^2 \leq e^{-\frac{\gamma^2}{4}}\|\theta^t - \theta_{opt}\|^2$$

This suggests that after $T = \frac{4}{\gamma^2}\ln(\frac{100}{\gamma^2})$ iterations where $\|\theta^t - \theta_{opt}\| \geq \gamma r_{opt}$ we get that

$$\|\theta^T - \theta_{opt}\|^2 \leq e^{-\frac{T\gamma^2}{4}}\|\theta_0 - \theta_{opt}\|^2 \leq \frac{\gamma^2}{100} \cdot 100 r_{opt}^2 = \gamma^2 r_{opt}^2$$

as required. Now, should it be the case that in some iteration $\|\theta^t - \theta_{opt}\| < \gamma r_{opt}$ and we make an update step. Again, Equation (1) asserts that

$$\|\theta^{t+1} - \theta_{opt}\|^2 \leq (1 - \frac{\gamma^2}{2})\|\theta^t - \theta_{opt}\|^2 + \frac{\gamma^4}{4} \cdot r_{opt}^2 < (1 - \frac{\gamma^2}{2})\gamma^2 r_{opt}^2 + \frac{\gamma^4}{4} \cdot r_{opt}^2 < \gamma^2 r_{opt}^2$$

so once $\|\theta^t - \theta_{opt}\| < \gamma r_{opt}$ then we have that $\|\theta^\tau - \theta_{opt}\| < \gamma r_{opt}$ for all $\tau \geq t$. $\qquad\square$

### 3.1 The Noisy/SQ-Version of the fPTAS for the MEB Problem

Now, we consider a scenario where in each iteration $t$, rather than using the exact mean $\mu_w^t = \frac{\sum_{x \in P \setminus B(\theta^t, r)} x}{|P \setminus B(\theta^t, r)|}$, we obtain an approximated mean $\tilde{\mu}_w^t = \mu_w^t + \Delta^t$. We consider here two scenarios: (a) where $\Delta^t$ is a zero-mean bounded-variance random noise — a setting we refer to as the *random noise* setting; and (b) where $\Delta^t$ is an arbitrary noise s.t. the constraint that $\|\Delta^t\| = O(\gamma r)$ — a setting we refer to as *arbitrary noise*. The latter isn't used in our algorithm and is deferred to Appendix B.

**The random noise setting.** In this setting, our update step in each iteration is made not using a deterministically chosen uncovered point but rather by a draw from a distribution $\mathcal{D}^t$ whose mean is "as good" as an uncovered point. This requires us to make two changes to the algorithm: (i) modify the update rate and (ii) repeat the entire algorithm $R = O(\log(1/\beta))$ times.

**Claim 3.4.** Consider an altered version of Algorithm 2 which (1) repeats the algorithm $R = \lceil\log_{4/3}(1/\beta)\rceil$ times, (2) each repetition is composed of at most $T = \frac{4096}{\gamma^2}\ln(\frac{121\cdot4}{\gamma^2})$ update-steps and (3) in each iteration where it doesn't terminate it draws a point $z \sim \mathcal{D}^t$ and makes that update-step: $\theta^{t+1} \leftarrow \theta^t + \frac{\gamma^2}{2048}z$. If it holds that for each iteration $t$ we have that $\mathcal{D}^t$ satisfies the two properties

$$\text{(i)} \quad \mathop{\mathbb{E}}_{z\sim\mathcal{D}^t}\left[\langle\theta_{opt} - \theta^t, z\rangle \mid \theta^t\right] \geq \frac{1}{4}\|\theta^t - \theta_{opt}\|^2$$

$$\text{(ii)} \quad \mathop{\mathbb{E}}_{z\sim\mathcal{D}^t}[\|z\|^2 \mid \theta^t] \leq 512r^2 \qquad (2)$$

then, provided that $r \geq r_{opt}$, we have that w.p. $\geq 1 - \beta$ one of the $R$ repetitions of the revised algorithm returns a candidate center $\theta^T$ where $P \subset B(\theta^T, (1 + \gamma)r)$.

*Proof.* To prove the claim it suffices to show that in a single execution of the algorithm we have that $\Pr[\|\theta^T - \theta_{opt}\| \leq \gamma r] = \Pr[\|\theta^T - \theta_{opt}\|^2 \leq \gamma^2 r^2] \geq 1/4$, implying that in $R$ repetitions of the algorithm the failure probability decreases to $(3/4)^R = \beta$. To that end, denote the non-negative random variables $Y^t = \|\theta^t - \theta_{opt}\|^2$ for each iteration $t$. Note that if we show that $\mathbb{E}[Y^T] \leq \frac{3}{4}\gamma^2 r^2$ then Markov's inequality implies that $\Pr[Y^T \geq \gamma^2 r^2] \leq 3/4$. So our goal is to prove that $\mathbb{E}[Y^T] \leq \frac{3}{4}\gamma^2 r^2$.

We can now analyze the conditional expectation and observe that

$$\mathbb{E}\left[\|\theta^{t+1} - \theta_{opt}\|^2 \mid \theta^t\right] = \mathbb{E}\left[\left\|\theta^t - \theta_{opt} + \frac{\gamma^2}{2048}z\right\|^2 \mid \theta^t\right]$$

$$= \mathbb{E}\left[\|\theta^t - \theta_{opt}\|^2 + \frac{2\gamma^2}{2048}\langle z, \theta^t - \theta_{opt}\rangle + (\frac{\gamma^2}{2048})^2\|z\|^2 \mid \theta^t\right]$$

$$\overset{z\sim\mathcal{D}^t}{\leq} \|\theta^t - \theta_{opt}\|^2 - \frac{2\gamma^2}{2048} \cdot \frac{1}{4}\|\theta^t - \theta_{opt}\|^2 + \frac{\gamma^4 \cdot 512 r^2}{2048^2} = (1 - \frac{\gamma^2}{4096})\|\theta^t - \theta_{opt}\|^2 + \frac{\gamma^4}{8192}r^2$$

Since $\mathbb{E}[Y^{t+1} \mid \theta^t] \leq (1 - \frac{\gamma^2}{4096})Y^t + \frac{\gamma^4}{8192}r^2$ then it is easy to see that $\mathbb{E}[Y^T] \leq (1 - \frac{\gamma^2}{4096})^T \cdot Y^0 + \frac{\gamma^4}{8192}r^2 \sum_{t=0}^{T-1}(1 - \frac{\gamma^2}{4096})^t \leq (1 - \frac{\gamma^2}{4096})^t \cdot (11r)^2 + \frac{\gamma^2}{2}r^2$. It follows that iteration $T = \frac{4096}{\gamma^2}\ln(\frac{121\cdot 4}{\gamma^2})$ we have that $\mathbb{E}[Y^T] \leq \frac{\gamma^2}{4}r^2 + \frac{\gamma^2}{2}r^2 = \frac{3}{4}\gamma^2 r^2$ as required. $\square$

**Corollary 3.5.** Suppose that in each iteration $t$ of the revised algorithm $\mathcal{D}^t$ is a distribution that satisfies the required two properties of Claim 3.4 w.p. $\geq 1 - \frac{1}{8T\cdot\lceil \log_{8/7}(1/\beta)\rceil}$. Then, repeating this algorithm $R = \lceil \log_{8/7}(1/\beta)\rceil$ many times we have that w.p. $\geq 1 - \beta$ it holds that for at least one repetition we have $P \subset (B(\theta^T, (1+\gamma)r)$.

*Proof.* Using the union bound, it follows that in one of the $R \cdot T$ repetition of the revised algorithm the probability that one draw isn't from a good $\mathcal{D}^t$ (that does satisfy these two properties) is at most $1/8$. It follows that $\Pr[Y^T \geq \gamma^2 r^2] \geq 1/4 - 1/8 = 1/8$. Repeating this algorithm $R$ reduces the failure probability to $(7/8)^R \leq \beta$. $\square$

# 4 A Differentially Private fPTAS for the MEB Problem

We now turn our attention to the privacy-preserving versions of Algorithms 1 and 2. In this section we give their curator-model $\rho$-zCDP versions (Algorithms 3 and 4 resp.), whereas in the following section (Section 5) we detail their local-model zCDP versions. Due to space constraints, (i) the proof that Algorithm 3 is $\rho$-zCDP, (ii) the the full proofs of the following two statements, and (iii) the application of Algorithm 3 to Subsample-and-Aggregate are all deferred to the Supplementary Material, Section C.

---

**Algorithm 3** Differentially Private Minimum Enclosing Ball (DP-MEB)

---

**Input:** a set of $n$ points $P \subseteq \mathbb{R}^d$, an approximation parameter $\gamma \in (0, 1)$,
an initial radius $r_0$ s.t. $r_{opt} \leq r_0 \leq 4r_{opt}$, and an initial center $\theta_0$ s.t. $\|\theta_0 - \theta_{opt}\| \leq 10r_{opt}$,
error parameter $\beta$ and privacy-parameter $\rho$.
1: Remove any $x \in P$ which doesn't belong to $B(\theta_0, 11r_0)$.
2: Set $i_{\min} \leftarrow 0$, $i_{\max} \leftarrow \ln_{1+\gamma}(4)(\approx \frac{4}{\gamma})$, and $\theta^* \leftarrow \theta_0$.
3: Set $B \leftarrow \lceil \log_2(\ln_{1+\gamma}(4))\rceil$.
4: **while** $(i_{\min} < i_{\max})$ **do**
5:     $i_{cur} = \lfloor \frac{i_{\min} + i_{\max}}{2}\rfloor$
6:     $r_{cur} \leftarrow (1+\gamma)^{i_{cur}} \cdot r_0/4$
7:     $\theta_{cur} \leftarrow$ DP-MMEB$(P, \gamma, \frac{\beta}{B}, \frac{\rho}{B}, r_{cur}, \theta_0)$
8:     **if** $(\theta_{cur} \neq \perp)$ **then**
9:         Set $i_{\max} \leftarrow i_{cur}$, $\theta^* \leftarrow \theta_{cur}$ and $r^* \leftarrow (1+\gamma)r_{cur}$
10:    **else**
11:        $i_{\min} \leftarrow i_{cur} + 1$
12: **return** $B(\theta^*, r^*)$

---

**Algorithm 4** DP-Margin based Minimum Enclosing Ball (DP-MMEB)

---

**Input:** a set of $n$ points $P \subseteq \mathbb{R}^d$, an approximation parameter $\gamma \in (0,1)$,
an error parameter $\beta \in (0,1)$, privacy parameter $\rho$,
a candidate radius $r$, and an initial center $\theta_0$ s.t. $\|\theta_0 - \theta_{opt}\| \leq 10 r_{opt}$.

1: Set $R \leftarrow \lceil \log_{8/7}(1/\beta) \rceil$, $\theta^0 \leftarrow \theta_0$, $T \leftarrow \frac{4096}{\gamma^2} \ln(\frac{484}{\gamma^2})$, $\beta_0 = \frac{1}{16RT}$, $\sigma_{count}^2 \leftarrow \frac{R(T+1)}{\rho}$, and
$\quad \sigma_{sum}^2 \leftarrow \frac{RT \cdot (88r)^2}{\rho}$.
2: **repeat**
3: $\quad$ **for** $t = 0, 1, 2, \ldots, T - 1$ **do**
4: $\quad\quad$ Sample $\Delta_{count} \sim \mathcal{N}(0, \sigma_{count}^2)$.
5: $\quad\quad$ $\tilde{n}_w^t \leftarrow |\{x \in P : x \notin B(\theta^t, r)\}| + \Delta_{count}$
6: $\quad\quad$ **if** $\left( \tilde{n}_w^t < \frac{88\sqrt{RT}}{\sqrt{\rho}} \left( \sqrt{d} + \sqrt{2 \ln(4RT/\beta_0)} \right) \right)$ **then**
7: $\quad\quad\quad$ **return** $\theta^t$
8: $\quad\quad$ Sample $\Delta_{sum} \sim \mathcal{N}(0, \sigma_{sum}^2 I_d)$.
9: $\quad\quad$ Set $\tilde{\mu}_w^t \leftarrow \frac{1}{\tilde{n}_{wt}} \left( \sum_{x \notin B(\theta^t, r)} (x - \theta^t) + \Delta_{sum} \right)$.
10: $\quad\quad$ Update $\theta^{t+1} \leftarrow \theta^t + \frac{\gamma^2}{2048} \tilde{\mu}_{wt}$
11: $\quad$ Sample $\Delta_{count} \sim \mathcal{N}(0, \sigma_{count}^2)$.
12: $\quad$ **if** $\left( |P \setminus B(\theta^T, (1+\gamma)r)\}| + \Delta_{count} \leq \sqrt{\frac{2R(T+1)\log(4R(T+1)/\beta_0)}{\rho}} \right)$ **then return** $\theta^T$ and
$\quad$ halt
13: **until** $R$ repetitions
14: **return** $\perp$

---

**Lemma 4.1.** W.p. $\geq 1 - \beta$, applying Algorithm 4 with $r \geq r_{opt}$ and an initial center $\theta_0$ s.t. $\|\theta_0 - \theta_{opt}\| \leq 10 r_{opt}$ returns a point $\theta^t$ where $|P \setminus B(\theta^t, (1+\gamma)r)| \leq 88\sqrt{\frac{RT}{\rho}} \left( \sqrt{d} + \sqrt{2 \ln(4RT/\beta_0)} \right) + \sqrt{\frac{2R(T+1)\log(4R(T+1)/\beta_0)}{\rho}}$.

*Proof.* Given a repetition $r$ and iteration $t$ denote the events

$$\mathcal{E}_1^{r,t} := \text{in the } (r,t)\text{-draws, } |\Delta_{count}| \leq \sigma_{count} \sqrt{2 \ln(4R(T+1)/\beta_0)}$$

$$\mathcal{E}_2^{r,t} := \text{in the } (r,t)\text{-draw, } \|\Delta_{sum}\| \leq \sigma_{sum} \left( \sqrt{d} + \sqrt{2 \ln(4RT/\beta_0)} \right)$$

and denote also $\mathcal{E}_i = \bigcup_{r,t} \mathcal{E}_i^{r,t}$ for $i = 1, 2$. Using standard bounds on the concentration of the Gaussian distribution and the $\chi_d^2$-distribution together with the union-bound we have that $\Pr[\overline{\mathcal{E}_1} \cup \overline{\mathcal{E}_2}] \leq R(T+1) \cdot \frac{\beta_0}{2R(T+1)} + RT \frac{\beta_0}{2RT} \leq \beta_0$.

Fix $r$ and $t$. Under $\mathcal{E}_1^{r,t} \cap \mathcal{E}_2^{r,t}$ holding, the required conditions detailed in (2) hold, which – using Corollary 3.5 – yields the correctness of our algorithm. Under the same notation as in Algorithm 4, denote the distribution of $\frac{1}{n_w^t + \Delta_{count}} \left( \sum_{x \notin B(\theta^t, r)} (x - \theta^t) + \Delta_{sum} \right)$ as $\mathcal{D}^t$.

First, observe that under $\mathcal{E}_1^{r,t}$, the condition $\tilde{n}_w^t \geq \frac{88\sqrt{RT}}{\sqrt{\rho}} \left( \sqrt{d} + \sqrt{2 \ln(4RT/\beta_0)} \right)$ implies that

$$n_w^t \geq \frac{88\sqrt{RT}}{\sqrt{\rho}} \left( \sqrt{d} + \sqrt{2 \ln(4RT/\beta_0)} \right) - \sqrt{\frac{2R(T+1)\ln(4R(T+1)/\beta_0)}{\rho}} \geq 44 |\Delta_{count}|$$

and secondly, observe that $\Delta_{sum}$ is drawn from a spherically symmetric distribution, so for any $a > 0$ we have that $\mathbb{E}[\Delta_{sum}| \|\Delta_{sum}\| \leq a] = 0$. And so, if indeed Algorithm 4 passes the if-condition and makes an update step we have

$$\mathbb{E}_{z \sim \mathcal{D}^t} \left[ \langle \theta_{opt} - \theta^t, z \rangle | \theta^t, \mathcal{E}_1^{r,t} \cap \mathcal{E}_2^{r,t} \right] = \left\langle \theta_{opt} - \theta^t, \mathbb{E} \left[ \frac{\Delta_{sum} + \sum_{x \notin B(\theta^t, r)} (x - \theta^t)}{n_w^t + \Delta_{count}} | \theta^t, \mathcal{E}_1^{r,t} \cap \mathcal{E}_2^{r,t} \right] \right\rangle$$

$$\overset{\text{independ.}}{=} \langle \theta_{opt} - \theta^t, \mathbb{E}\left[\frac{1}{n_w^t + \Delta_{count}}\Big| \theta^t, \mathcal{E}_1^{r,t}\right] \sum_{x \notin B(\theta^t, r)} (x - \theta^t)\rangle$$

$$= \langle \theta_{opt} - \theta^t, \mathbb{E}\left[\frac{n_w^t}{n_w^t + \Delta_{count}}\Big| \theta^t, \mathcal{E}_1^{r,t}\right] \frac{\sum_{x \notin B(\theta^t, r)} (x - \theta^t)}{n_w^t}\rangle$$

$$= \mathbb{E}\left[\frac{1}{1 + \Delta_{count}/n_w^t}\Big| \theta^t, \mathcal{E}_1^{r,t}\right] \langle \theta_{opt} - \theta^t, \frac{\sum_{x \notin B(\theta^t, r)} x}{n_w^t} - \theta^t\rangle \overset{\text{Clm 3.3}}{\geq} \left(\frac{1}{1 - 1/44}\right) \cdot \frac{1}{2}\|\theta^t - \theta_{opt}\|^2 \geq \frac{1}{4}\|\theta^t - \theta_{opt}\|^2$$

and also

$$\mathbb{E}_{z \sim \mathcal{D}^t}[\|z\|^2 | \theta^t, \mathcal{E}_1^{r,t} \cap \mathcal{E}_2^{r,t}] = \mathbb{E}\left[\left\|\frac{\sum_{x \notin B(\theta^t, r)} (x - \theta^t)}{n_w^t + \Delta_{count}} + \frac{\Delta_{sum}}{n_w^t + \Delta_{count}}\right\|^2 \Big| \theta^t, \mathcal{E}_1^{r,t} \cap \mathcal{E}_2^{r,t}\right]$$

$$= \mathbb{E}\left[\left(\frac{n_w^t}{n_w^t + \Delta_{count}}\right)^2 \left\|\frac{\sum_{x \notin B(\theta^t, r)} x}{n_w^t} - \theta^t\right\|^2 + \frac{2\langle\Delta_{sum}, \sum_{x \notin B(\theta^t, r)}(x - \theta^t)\rangle + \|\Delta_{sum}\|^2}{(n_w^t + \Delta_{count})^2}\Big| \theta^t, \mathcal{E}_1^{r,t} \cap \mathcal{E}_2^{r,t}\right]$$

$$\overset{\text{independ.}}{=} \mathbb{E}\left[\left(\frac{1}{1 + \frac{\Delta_{count}}{n_w^t}}\right)^2\Big| \theta^t, \mathcal{E}_1^{r,t}\right] \left\|\frac{\sum_{x \notin B(\theta^t, r)} x}{n_w^t} - \theta^t\right\|^2 + \frac{0 + \mathbb{E}\left[\|\Delta_{sum}\|^2 | \theta^t, \mathcal{E}_2^{r,t}\right]}{(\tilde{n}_w^t)^2}$$

$$\leq \frac{1}{1 - 1/44} \cdot (11r)^2 + \frac{RT \cdot (88r)^2}{\rho}\left(\sqrt{d} + \sqrt{2\ln(4RT/\beta_0)}\right)^2 \cdot \frac{1}{(\tilde{n}_w^t)^2} < 512r^2$$

since $\tilde{n}_w^t \geq 88\sqrt{\frac{RT}{\rho}}\left(\sqrt{d} + \sqrt{2\ln(4RT/\beta_0)}\right)$ in order for us to make an update.

Corollary 3.5 suggests that if we make all $T$ updates then indeed $\|\theta^T - \theta_0\| \leq \gamma r$ and so $|P \setminus B(\theta^T, (1 + \gamma)R)| = 0$. So under $\mathcal{E}_1$ Algorithm 4 returns $\theta^T$. Otherwise, at some iteration we do not make an update step, which under $\mathcal{E}_1$ suggests that

$$n_w^t = |P \setminus B(\theta^t, r)| \leq 88\sqrt{\frac{RT}{\rho}}\left(\sqrt{d} + \sqrt{2\ln(4RT/\beta_0)}\right) + \sqrt{\frac{2R(T+1)\log(4R(T+1)/\beta_0)}{\rho}} \quad \square$$

**Corollary 4.2.** Given $r_0$ where $r_{opt} \leq r_0 \leq 4r_{opt}$ and a point $\theta_0$ where $\|\theta_0 - \theta^*\| \leq 10r_{opt}$, w.p. $\geq 1 - \beta$ Algorithm 3 is a $O(n \cdot \frac{\log^2(1/\gamma)\log(1/\beta)}{\gamma^2})$-time algorithm that returns a ball $B(\theta^*, r)$ where

$$r \leq (1 + 3\gamma)r_{opt} \text{ and where } |P \setminus B(\theta^*, r^*)| = O(\frac{\left(\sqrt{d} + \sqrt{\log(\log(1/\beta)/\gamma)}\right)\sqrt{\log(1/\gamma)\log(1/\beta)}}{\gamma\sqrt{\rho}}).$$

The proof is deferred to Supplementary Material, Section C. We comment that the amplification of the success probability of the algorithm from $1/8$ to $1 - \beta$ can be done using the amplification techniques of [28] which saves on the privacy budget: instead of naïvely setting the privacy budget per iteration as $\rho/R$, we could use conversions to $(\epsilon, \delta)$-DP and as a result "shave-off" a factor of $R$. But since $R = O(\log(1/\beta))$ this would merely reduce polyloglog factors, at the expense of readability.

## 5 A Local-DP fPTAS for the MEB Problem

In this section we give the local-model version of our algorithm. At the core of its utility proof is a lemma analogous to Lemma 4.1, in which we prove that w.h.p. in each iteration $t$ the distribution of our update-step satisfies (w.h.p.) the requirements of (2). Again, due to space constraints, we merely state the LDP algorithm in this section, and defer both its privacy and utility analyses to the Supplementary Material, Section D — where we prove that it is a $\rho$-zCDP algorithm that returns a ball $B(\theta^*, r^*)$ such that $r^* \leq (1 + 3\gamma)r_{opt}$ and $|P \setminus B(\theta^*, r^*)| = O(\frac{\sqrt{n}\log(1/\gamma)}{\gamma^2\sqrt{\rho}}\left(\sqrt{d} + \sqrt{\log(1/\gamma\beta)}\right))$.

---
**Algorithm 5** LDP-Margin based Minimum Enclosing Ball (LDP-MMEB)
___

**Input:** a set of $n$ points $P \subseteq \mathbb{R}^d$, an approximation parameter $\gamma \in (0, 1)$,
an error parameter $\beta \in (0, 1)$, privacy parameter $\rho$,
a candidate radius $r$, and an initial center $\theta_0$ s.t. $\|\theta_0 - \theta_{opt}\| \leq 10 r_{opt}$.

1: Set $R \leftarrow \lceil \log_{8/7}(1/\beta) \rceil$, $\theta^0 \leftarrow \theta_0$, $T \leftarrow \frac{4096}{\gamma^2} \ln(\frac{484}{\gamma^2})$, $\beta_0 = \frac{1}{16RT}$, $\sigma^2_{count} \leftarrow \frac{R(T+1)}{\rho}$, and
$\quad \sigma^2_{sum} \leftarrow \frac{RT \cdot (88r)^2}{\rho}$.

2: **repeat**

3:     **for** $t = 0, 1, 2, \ldots, T - 1$ **do**

4:         **for each** $(x \in P)$ **do**

5:             Sample $\Delta_{count} \sim \mathcal{N}(0, \sigma^2_{count})$.

6:             Sample $\Delta_{sum} \sim \mathcal{N}(0, \sigma^2_{sum} I_d)$.

7:             **if** $(x \notin B(\theta^t, r))$ **then**

8:                 Send $Y^t_x = 1 + \Delta_{count}$, $Z^t_x = x - \theta^t + \Delta_{sum}$

9:             **else** Send $Y^t_x = \Delta_{count}$, $Z^t_x = \Delta_{sum}$

10:         Set $\tilde{n}^t_w = \sum_{x \in P} Y^t_x$ and $\tilde{v}^t_w = \frac{1}{\tilde{n}^t_w} \sum_{x \in P} Z^t_x$.

11:         **if** $\left( \tilde{n}^t_w < \frac{88\sqrt{nRT}}{\sqrt{\rho}} \left( \sqrt{d} + \sqrt{2\ln(4RT/\beta_0)} \right) \right)$ **then return** $\theta^t$

12:         Update $\theta^{t+1} \leftarrow \theta^t + \frac{\gamma^2}{2048} \tilde{v}^t_w$

13:     **for each** $(x \in P)$ **do**

14:         Sample $\Delta_{count} \sim \mathcal{N}(0, \sigma^2_{count})$.

15:         **if** $(x \notin B(\theta^T, (1 + \gamma)r))$ **then**

16:             Send $Y^T_x = 1 + \Delta_{count}$

17:         **else** Send $Y^T_x = \Delta_{count}$

18:     Set $n^T_w \leftarrow \sum_x Y_x$

19:     **if** $\left( n^T_w \leq \sqrt{\frac{2nR(T+1)\log(4R(T+1)/\beta_0)}{\rho}} \right)$ **then return** $\theta^T$ and halt

20: **until** $R$ repetitions

21: **return** $\perp$
___

## 6 Discussion and Open Problems

This work is the first to give a DP-fPATS for the MEB problem, in both the curator- and the local-model, and it leads to numerous open problems. The first is the question of improving the utility guarantee. Specifically, the number of points our algorithm may omit from $P$ has a dependency of $\tilde{O}(1/\gamma)$ in the approximation factor, where this dependency follows from the fact that in each of our $T = \tilde{O}(\gamma^{-2})$ iterations. Thus finding either an iterative algorithm which makes $\ll T$ iterations or a variant of SVT that will allow the privacy budget to scale like $O(\log(T))$ will reduce this dependency to only $\text{polylog}(\gamma^{-1})$. Alternatively, it is intriguing whether there exists a lower-bound for any zCDP PTAS of the MEB problem proving a polynomial dependency on $\gamma$. (The best we were able to prove is via packing argument [23, 8] using a grid of $O((1/\gamma)^d)$ many points, leading to a $d \log(1/\gamma)$ bound.)

A different open problem lies on the the application of this DP-MEB approximation to the task of DP-clustering, and in particular — on improving on the works of [24, 34, 14] for "stable" $k$-median/means clustering. One can presumably combine our technique with the LSH-based approach used in [31] to cover a subset of points lying close together, however — it is unclear to us what is the effect of using only some of each cluster's "core" on the approximated MEB we return and on the $k$-means/median cost. But it is possible that our work can be a building block in a first PTAS for the $k$-center problem in low-dimensions, a setting in which $k$-center has a non-private PTAS [22].

## Acknowledgments and Disclosure of Funding

O.S. is supported by the BIU Center for Research in Applied Cryptography and Cyber Security in conjunction with the Israel National Cyber Bureau in the Prime Ministers Office, and by ISF grant no. 2559/20. Both authors thank the anonymous reviewers for terrific suggestions and advice on improving this paper.

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
