# A Finding an Initial Good Center

In this section we give, for completeness, the $\rho$-zCDP version of the algorithms for approximating $P$'s optimal radius up to a constant factor and finding some $\theta_0$ which is sufficiently close to the center of $P$'s MEB. The algorithm itself is ridiculously simple, and has appeared before implicitly. We bring it here for two reasons: (a) completeness and (b) in its LDP-version, this algorithm's utility depends solely on $\sqrt{n}$. Thus, combining this algorithm with the Algorithm 5 of Section 5, we obtain a LDP-fPTAS for the MEB problem who's utility depends on $\sqrt{n}$ rather than the $n^{0.67}$-bound of [31] (at the expense of worse dependency on other parameters). This gives a clear improvement on previous algorithms for approximating the MEB problem when $n \to \infty$. Our algorithm requires a starting point $\theta_0$ which is $R_{\max}$ away from all points in $P$ (namely, $P \subset B(\theta_0, R_{\max})$, and a lower bound $r_{\min}$ on $r_{opt}$; and its overall utility bounds depends on $\log(R_{\max}/r_{\min})$. In a standard setting, where $P \subset [-B, B]^d$ and where all points lie on some grid $\mathcal{G}^d$ whose step-size is $\tau$, we can set $\theta_0$ as the origin and set $R_{\max} = B\sqrt{d}$ and $r_{\min} = \tau/2$, resulting in $O(\log(Bd/\tau))$-dependency. In the specific case where $r_{opt} = 0$ and all datapoints in $P$ lie on the exact same grid point we can just return the closest grid point to the resulting $\theta$ once it get to a radius of $r = r_{min} = \tau/2$.

---

**Algorithm 6** Noisy Average and Radius (GoodCenter)

---

    **Input:** a set of $n$ points $P$ and parameters $\theta_0, R_{\max}$ and $r_{\min}$, such that $P \subset B(\theta_0, R_{\max})$ and $r_{opt} \geq r_{\min}$. Failure parameter $\beta \in (0, 1)$, privacy parameter $\rho$.

1: Set $T \leftarrow \lceil \log_2(R_{\max}/r_{\min}) \rceil + 1$, $X \leftarrow \sqrt{\frac{2T \ln(4T/\beta)}{\rho}}$
2: Set $\sigma_{count}^2 \leftarrow \frac{T}{\rho}, \sigma_{sum}^2 \leftarrow \frac{T}{\rho}$.
3: Init $P^0 \leftarrow P, \theta^0 \leftarrow \theta_0, n_{cur} \leftarrow n$ and $r_{cur} \leftarrow R_{\max}$.
4: **for** $(t = 0, 1, 2, ..., T - 1)$ **do**
5:     $P^t \leftarrow P^t \cap B(\theta^t, r_{cur})$.
6:     $\Delta_{sum} \sim \mathcal{N}(0, 4r_{cur}^2 \sigma_{sum}^2 I_d)$
7:     $\tilde{\mu}^t \leftarrow (\sum_{x \in P^t} x + \Delta_{sum})/n_{cur}$
8:     $\Delta_{count} \leftarrow \mathcal{N}(0, \sigma_{count}^2)$
9:     **if** $(|P^t \setminus B(\tilde{\mu}^t, \frac{1}{2} r_{cur})| + \Delta_{count} \geq X)$ **then return** $B(\theta^t, r_{cur})$
10:    Update: $r_{cur} \leftarrow \frac{1}{2} r_{cur}, n_{cur} \leftarrow n_{cur} - 2X, \theta^{t+1} \leftarrow \tilde{\mu}^t$.
11: **return** $B(\theta^T, r_{cur})$

---

**Theorem A.1.** *Algorithm 6 is $\rho$-zCDP.*

*Proof.* The proof follows immediately from the fact that the $L_2$-global sensitivity of a count query is 1, and that the $L_2$-global sensitivity of a sum of datapoints in a ball of radius $r_{cur}$ is at most $2r_{cur}$. The rest of the proof relies on the composition of $2T$ queries, each answered with a "budget" of $\frac{\rho}{2T}$-zCDP. $\qquad\square$

**Theorem A.2.** *W.p.* $\geq 1 - \beta$, *given a set of points $P$ of size $n$ where $n \geq \max\{16T\sqrt{\frac{2T \ln(4T/\beta)}{\rho}}, 16\sqrt{\frac{T}{\rho}}(\sqrt{d} + \sqrt{2\ln(4T/\beta)})\}$, Algorithm 6 returns a ball $B(\theta^*, r^*)$ where (i) the set $P' = P \cap B(\theta^*, r^*)$ contains at least $n - \sqrt{\frac{8T^3 \ln(4T/\beta)}{\rho}}$, and (ii) denoting $B(\theta(P'), r_{opt}(P'))$ as the MEB of $P'$, we have that $r^* \leq 6r_{opt}$.*

*Proof.* Let $\mathcal{E}$ be the event where for any of the $\leq T$ draws of the $\Delta_{sum}$ and $\Delta_{count}$ it holds that

$$|\Delta_{count}| \leq \sqrt{\frac{2T \ln(4T/\beta)}{\rho}} \qquad \text{and} \qquad \|\Delta_{sum}\| \leq 2r_{cur}\sqrt{\frac{T}{\rho}}(\sqrt{d} + \sqrt{2\ln(4T/\beta)})$$

where again, standard union bound and Gaussian / $\chi^2$-distribution concentration bounds give that $\Pr[\overline{\mathcal{E}}] \leq \beta$. So we continue the proof under the assumption that $\mathcal{E}$ holds.

In this case, in any iteration it must hold that $|P \setminus B(\mu^t, \frac{1}{2} r_{cur})| \leq 2X = \sqrt{\frac{8T \ln(4T/\beta)}{\rho}}$. It follows that all in all we remove in the process of Algorithm 6 at most $2XT$ points, and since $n \geq 16XT$

we have that in any iteration $t$ it always holds that $n \geq |P^t| \geq n - 2Xt = n_{cur} \geq \frac{7n}{8} \geq 14XT$. Denoting in any iteration $t$ the true mean of the points (remaining) in $P^t$ as $\mu_t = \frac{1}{|P^t|}\sum_{x \in P^t} x$, and the center of the MED of $P^t$ as $\theta_t$, it follows that

$$\|\tilde{\mu}^t - \mu^t\| = \|\tilde{\mu}^t - \theta_t - (\mu^t - \theta_t)\| = \left\| \frac{\Delta_{sum} + \sum_{x \in P^t}(x - \theta_t)}{n_{cur}} - \frac{\sum_{x \in P^t}(x - \theta_t)}{|P^t|} \right\|$$

$$\leq \left\| \frac{\Delta_{sum}}{n_{cur}} \right\| + \left\| \frac{\left(\sum_{x \in P^t}(x - \theta_t)\right)(|P^t| - n_{cur})}{|P^t|n_{cur}} \right\| \leq \frac{8\|\Delta_{sum}\|}{7n} + \|\mu^t - \theta_t\|\frac{2XT}{n_{cur}}$$

$$\leq \frac{8 \cdot 2r_{cur}\sqrt{\frac{T}{\rho}}(\sqrt{d} + \sqrt{2\ln(4T/\beta)})}{7n} + \frac{r_{opt}(P^t)}{7} \leq \frac{r_{cur} + r_{opt}(P^t)}{7}$$

Since we assume $n \geq 16\sqrt{\frac{T}{\rho}}(\sqrt{d} + \sqrt{2\ln(4T/\beta)})$. Moreover, since $\|\mu^t - \theta_t\| \leq r_{opt}(P^t)$ it follows that $\|\tilde{\mu}^t - \theta_t\| \leq \frac{r_{cur} + 8r_{opt}(P^t)}{7}$. Now, as long as $r_{cur} \geq 6r_{opt}(P^t)$ we have that

$$\frac{r_{cur}}{2} \geq \frac{r_{cur}}{7} + \frac{5r_{cur}}{14} \geq \frac{r_{cur}}{7} + \frac{30r_{opt}(P^t)}{14} \geq r_{opt}(P^t) + \frac{r_{cur} + 8r_{opt}(P^t)}{7} \geq r_{opt}(P^t) + \|\tilde{\mu}^t - \theta_t\|$$

thus $B(\theta_t, r_{opt}(P^t)) \subset B(\tilde{\mu}^t, \frac{r_{cur}}{2})$ which implies that $|P^t \setminus B(\mu^t, \frac{1}{2}r_{cur})| = 0$, and so under $\mathcal{E}$ we continue to the next iteration.

And so, when we halt it must hold that $r_{cur}$ (which is the $r^*$ we return) must satisfy that $r_{cur} < 6r_{opt}(P^t)$. $\qquad\square$

**Corollary A.3.** Algorithm 6 is a $\rho$-zCDP algorithm that, given $n$ points on a grid $\mathcal{G} \subset [-B, B]^d$ of side-step $\tau$ where $n = \Omega(\sqrt{\frac{\log(Bd/\tau)}{\rho}}(\sqrt{d} + \sqrt{\log(Bd/\tau\beta)}))$ returns w.p. $\geq 1 - \beta$ a ball $B(\theta^*, r^*)$ where for $P' = P \setminus B(\theta^*, r^*)$ it holds that both $n - |P'| = O(\frac{\log(Bd/\tau)}{\sqrt{\rho}}\sqrt{\log(Bd/\tau\beta)})$ and that w.r.t to $B(\theta_{opt}, r_{opt})$ which is the true MEB of $P'$ we have that $\|\theta^* - \theta_{opt}\| \leq 6r_{opt}(P')$.

### A.1 A Local-DP Version of Finding an Initial Good Center

---

**Algorithm 7** LDP Noisy Average and Radius (LDP-GoodCenter)

---

**Input:** a set of $n$ points $P$ and some parameter $R_{\max}, \theta_0$ and $r_{\min}$, such that $P \subset B(\theta_0, R_{\max})$ and $r_{opt} \geq r_{\min}$. Failure parameter $\beta \in (0, 1)$, privacy parameter $\rho$.

1: Set $T \leftarrow \lceil \log_2(R_{\max}/r_{\min}) \rceil + 1$, $X \leftarrow \sqrt{\frac{2nT \ln(4T/\beta)}{\rho}}$
2: $\sigma^2_{count} \leftarrow \frac{T}{\rho}$, $\sigma^2_{sum} \leftarrow \frac{T}{\rho}$.
3: Init $\theta^0 \leftarrow \theta_0$, and $r_{cur} \leftarrow R_{\max}$.
4: **for** $(t = 0, 1, 2, ..., T - 1)$ **do**
5:     Denote $\Pi^t$ as the projection onto $B(\theta^t, r_{cur})$.
6:     **for each** $x \in P$ **do**
7:         Send $Y_x \sim \mathcal{N}(\Pi^t(x), 4r^2_{cur}\sigma^2_{sum}I_d)$
8:     $\tilde{\mu}^t \leftarrow \frac{1}{n}\sum_x Y_x$
9:     **for each** $x \in P$ **do**
10:        **if** $(x \notin B(\tilde{\mu}^t, \frac{1}{2}r_{cur}))$ **then**
11:            Send $Z_x \sim \mathcal{N}(1, \sigma^2_{count})$
12:        **else** Send $Z_x \sim \mathcal{N}(0, \sigma^2_{count})$
13:     **if** $(\sum_x Z_x \geq X)$ **then return** $B(\theta^t, r_{cur})$
14:     Update: $r_{cur} \leftarrow \frac{1}{2}r_{cur}$, $\theta^{t+1} \leftarrow \tilde{\mu}^t$.
15: **return** $B(\theta^T, r_{cur})$

---

**Theorem A.4.** *Algorithm 7 is a LDP algorithm in which each user maintains $\rho$-zCDP. Forthermore, w.p. $\geq 1 - \beta$, given a set of point $P$ of size $n$ where $n \geq \max\{16T\sqrt{\frac{2nT\ln(4T/\beta)}{\rho}}, 16\sqrt{\frac{nT}{\rho}}(\sqrt{d} +$*

$\sqrt{2\ln(4T/\beta)})\}$, *Algorithm 7 returns a ball* $B(\theta^*, r^*)$ *where the set* $P' = \{\Pi_{B(\theta^*, r^*)}(x) : x \in P\}$ *contains no more than* $2T\sqrt{\frac{2T\ln(4T/\beta)}{\rho}}$ *points for which* $x \neq \Pi_{B(\theta^*, r^*)}(x)$; *and denoting* $B(\theta(P'), r_{opt}(P'))$ *as the MEB of* $P'$, *it holds that* $\|\theta^* - \theta(P')\| \leq 8r*$.

The proof of Theorem A.4 is completely analogous to the proof of Theorems A.1 and A.2 using the fact that in each iteration $t$ of the algorithm

$$\sum_x Y_x \sim \mathcal{N}\left(\sum_x \Pi^t(x), \ 4nr_{cur}^2\sigma_{sum}^2 I_d\right)$$

$$\sum_x Z_x \sim \mathcal{N}\left(|\{x \in P : x \notin B(\tilde{\mu}^t, r_{cur}/2)\}|, \ n\sigma_{count}^2\right)$$

**Corollary A.5.** Algorithm 7 is a $\rho$-zCDP algorithm in the local-model that, given $n$ points on a grid $\mathcal{G} \subset [-B, B]^d$ of side-step $\tau$ where $n = \Omega(\frac{\log(Bd/\tau)}{\rho}(\sqrt{d} + \sqrt{\log(Bd/\tau\beta)})^2)$ returns w.p. $\geq 1 - \beta$ a ball $B(\theta^*, r^*)$ where for the set $P' = \{\Pi_{B(\theta^*, r^*)}(x) : x \in P\}$ it holds that at most $O(\frac{\sqrt{n}\cdot\log(Bd/\tau)}{\sqrt{\rho}}\sqrt{\log(Bd/\tau\beta)})$ points are shifted in the projection (and the rest remain as they are in $P$) and that w.r.t to $B(\theta_{opt}, r_{opt})$ which is the true MEB of $P'$ we have that $\|\theta^* - \theta_{opt}\| \leq 6r^*$.

Note that comparing Corollary A.5 with the approximation of [31], we have that they may omit $O(n^{0.67}\log(n/\tau))$-many points whereas we may omit only $\sqrt{n}\log^{3/2}(d/\tau)$ points. But, of course, they deal with a bounding ball for $t$ points out of giving $n$, whereas we deal with the MEB problem.

# B   Using Noisy Mean

Here we continue the analysis detailed in Section 3.1. For completeness, we also bring the SQ-model version of the algorithm where in each iteration we obtain an approximated center $\tilde{\mu}^t$ where $\Delta^t = \tilde{\mu}_w^t - \mu_w^t$ is of magnitude propostional to $\gamma r$. We modify Algorithm 2 so that our update scale shrinks by a constant factor to $\gamma^2/8$, namely we set $\theta^{t+1} \leftarrow (1 - \frac{\gamma^2}{8})\theta^t + \frac{\gamma^2}{8}\tilde{\mu}_w^t$. We now prove that the revised algorithm still converges to a point close to $\theta_{opt}$.

**Lemma B.1.** Applying Algorithm 2 with any $4r_{opt} \geq r \geq r_{opt}$ and any $\theta_0$ where $\|\theta_0 - \theta_{opt}\| \leq 10r_{opt}$, where in each iteration we use an approximated mean $\tilde{\mu}_w^t = \mu_w^t + \Delta^t$ where $\|\Delta^t\| \leq \frac{\gamma r}{16} \leq \frac{\gamma r_{opt}}{4}$ we obtain a $\theta$ where $\|\theta - \theta_{opt}\| \leq \gamma r_{opt}$ in at most $16T = \frac{64}{\gamma^2}\ln(100/\gamma^2)$ iterations.

*Proof.* First, analogously to Lemma 3.2 we have that in each update step we get

$$\|\theta^{t+1} - \theta_{opt}\|^2 = \left\|\left((1 - \frac{\gamma^2}{8})\theta^t + \frac{\gamma^2}{8}\tilde{\mu}_w^t\right) - \theta_{opt}\right\|^2 = (1 - \frac{\gamma^2}{8})^2 \cdot \|\theta^t - \theta_{opt}\|^2$$

$$+ 2\frac{\gamma^2}{8}(1 - \frac{\gamma^2}{8})\left(\langle\theta^t - \theta_{opt}, \mu_w^t - \theta_{opt}\rangle + \langle\theta^t - \theta_{opt}, \Delta^t\rangle\right) + (\frac{\gamma^2}{8})^2 \cdot \|\mu_w^t - \theta_{opt} + \Delta^t\|^2$$

$$\leq (1 - \frac{\gamma^2}{8})^2 \cdot \|\theta^t - \theta_{opt}\|^2 + 2(\frac{\gamma^2}{8} - \frac{\gamma^4}{64}) \cdot \left(\frac{1}{2}\|\theta^t - \theta_{opt}\|^2 + \|\theta^t - \theta_{opt}\| \cdot \frac{\gamma r_{opt}}{4}\right)$$

$$+ (\frac{\gamma^2}{8})^2 \cdot \left(2\|\mu_w^t - \theta_{opt}\|^2 + 2\frac{\gamma^2 r_{opt}^2}{4^2}\right)$$

$$\leq (1 - \frac{\gamma^2}{8})^2\|\theta^t - \theta_{opt}\|^2 + 2(\frac{\gamma^2}{8} - \frac{\gamma^4}{64}) \cdot \|\theta^t - \theta_{opt}\|\left(\frac{1}{2}\|\theta^t - \theta_{opt}\| + \frac{\gamma r_{opt}}{4}\right) + \frac{3\gamma^4}{64}r_{opt}^2$$

It follows that in each iteration where $\|\theta^t - \theta_{opt}\| \geq \gamma r_{opt}$ we get that

$$\|\theta^{t+1} - \theta_{opt}\|^2 \leq (1 - \frac{2\gamma^2}{8} + \frac{\gamma^4}{64})\|\theta^t - \theta_{opt}\|^2 + 2(\frac{\gamma^2}{8} - \frac{\gamma^4}{64}) \cdot \frac{3}{4}\|\theta - \theta_{opt}\|^2 + \frac{3\gamma^4 r_{opt}^2}{64}$$

$$< (1 - \frac{\gamma^2}{16})\|\theta^t - \theta_{opt}\|^2 + \frac{3\gamma^2}{64}\|\theta^t - \theta_{opt}\|^2 = (1 - \frac{\gamma^2}{64})\|\theta^t - \theta_{opt}\|^2$$

suggesting that after $16T = \frac{64}{\gamma^2}\ln(100/\gamma^2)$ iteration at most it must hold that

$$\|\theta^{16T} - \theta_{opt}\|^2 \le \exp(-\frac{64}{\gamma^2}\ln(100/\gamma^2) \cdot \frac{\gamma^2}{64})\|\theta_0 - \theta_{opt}\|^2 \le \frac{\gamma^2}{100} \cdot 100 r_{opt}^2 = \gamma^2 r_{opt}^2$$

As required. Similarly, if at some iteration $t$ it holds that $\|\theta^t - \theta_{opt}\| < \gamma r_{opt}$ then we get that

$$\|\theta^{t+1} - \theta_{opt}\|^2 \le (1 - \frac{\gamma^2}{8})^2 \gamma^2 r_{opt}^2 + 2(\frac{\gamma^2}{8} - \frac{\gamma^4}{64}) \cdot \frac{3}{4}\gamma^2 r_{opt}^2 + \frac{3\gamma^4 r_{opt2}^2}{64}$$

$$\le \gamma^2 r_{opt}^2 \left( 1 - \frac{2\gamma^2}{8} + \frac{\gamma^4}{64} + \frac{3\gamma^2}{2 \cdot 8} - \frac{3\gamma^4}{2 \cdot 64} + \frac{3\gamma^2}{64} \right) \le (1 - \frac{\gamma^2}{64})\gamma^2 r_{opt}^2$$

suggesting yet again that $\|\theta^\tau - \theta_{opt}\| < \gamma r_{opt}$ for all $\tau \ge t$. $\qquad\square$

## C  Missing Proofs: DP Algorithm

### C.1  Privacy Analysis

**Lemma C.1.** Algorithm 4 satisfies $\rho$-zCDP.

*Proof.* At each one of the $RT$ iterations of the algorithm, we answer two queries to the input data: a counting query and a summation query. It is known that the $L_2$-*sensitivity* of a counting query is 1, therefore using the Gaussian mechanism theorem while setting $\sigma_{count}^2 = \frac{R(T+1)}{\rho}$ satisfies $\frac{\rho}{2R(T+1)}$-zCDP. Secondly, we know that all the points are bounded by a ball of radius $11r_0 \le 44r_{opt} \le 44r$ around $\theta_0$, hence the summation query has $L_2$-sensitivity of $\le 88r$. Thus, by setting $\sigma_{sum}^2 = \frac{RT(88r)^2}{\rho}$ we have that we answer each summation query using $\frac{\rho}{2T}$-zCDP. Due to sequential composition of zCDP [9], it holds that in all $T$ iteration together we preserve $\left(\rho(1 - \frac{1}{2R(T+1)})\right)$-zCDP. Lastly, we apply one last counting query which we answer using the Gaussian mechanism while satisfying $\frac{\rho}{2R(T+1)}$-zCDP, thus, overall we are $\rho$-zCDP. $\qquad\square$

**Corollary C.2.** Algorithm 3 satisfies $\rho$-zCDP.

*Proof.* Since Algorithm 3 invokes $B = \lceil \log_2(\log_{1+\gamma}(4)) \rceil$ calls to Algorithm 4 each preserving $\frac{\rho}{B}$-zCDP, Algorithm 3 is $\rho$-zCDP overall. $\qquad\square$

### C.2  Utility Analysis and Sample Complexity

**Corollary C.3.** [Corollary 4.2 restated] Given $r_0$ where $r_{opt} \le r_0 \le 4r_{opt}$ and a point $\theta_0$ where $\|\theta_0 - \theta^*\| \le 10r_{opt}$, w.p. $\ge 1 - \beta$ Algorithm 3 is a $O(n \cdot \frac{\log^2(1/\gamma)\log(1/\beta)}{\gamma^2})$-time algorithm that returns a ball $B(\theta^*, r)$ where $r \le (1 + 3\gamma)r_{opt}$ and where $|P \setminus B(\theta^*, r^*)| = O(\frac{\left(\sqrt{d} + \sqrt{\log(\log(1/\beta)/\gamma)}\right)\sqrt{\log(1/\gamma)\log(1/\beta)}}{\gamma\sqrt{\rho}})$.

*Proof.* The result follows directly from the fact that Algorithm 3 invokes $B = O(\log(1/\gamma))$ calls to Algorithm 4, with a privacy budget of $O(\rho/\log(1/\gamma))$ each and with a failure probability of $O(\beta/\log(1/\gamma))$ each. Plugging those into the bound of Lemma 4.1 together with the fact that $T = O(\gamma^{-2}\log(1/\gamma))$ yields the resulting bound. Note that, denoting the "correct" $i^* = \min\{i \ge 0 : \frac{r_0}{4}(1 + \gamma)^i \ge r_{opt}\}$, under the event that no invocation of Algorithm 4 fails, each time we execute the binary search with a value of $i_{cur} \ge i^*$ we obtain some $\theta_{cur} \ne \perp$. Due to the nature of the binary search and the fact that upon finding $\theta_{cur} \ne \perp$ we set $i_{max} = i_{cur}$, it must follows that we return a ball of radius $(1 + \gamma)r^* = (1 + \gamma) \cdot \frac{r_0}{4} \cdot (1 + \gamma)^i$ for some $i \le i^*$, and so $r^* \le (1 + \gamma)^2 r_{opt} \le (1 + 3\gamma)r_{opt}$. Lastly, the runtime of Algorithm 4 is $O(nRT)$ making the runtime of Algorithm 3 to be $O(nRTB) = O(\frac{n\log^2(1/\gamma)\log(1/\beta)}{\gamma^2})$ as required. $\qquad\square$

## C.3 Application: Subsample Stable Functions

Much like the work of [21], our work too is applicable as a DP-aggregator in a Subsample-and-Aggregate [30] framework. We say that a point $p \in \mathbb{R}^d$ is $(r, \beta)$-*stable* for some function $f : \mathcal{X}^* \to \mathbb{R}^d$ if there exists $m(r, \beta)$ such that for any input $S \subset \mathcal{X}^n$ a random subsample of $m$ entries of $S$ input datapoints returns w.p. $\geq 1 - \beta$ a value close to $p$, namely, $\Pr_{S' \subset S, |S|=m}[\|c - f(S')\| \leq r] \geq 1 - \beta$.

**Theorem C.4.** *Fix $\rho, \gamma, \beta > 0$. There exists some constant $C > 0$ such that the following holds. Suppose $f : \mathcal{X}^* \to \mathbb{R}^d$ is a function that has a $(r, \beta)$-stable point. Then, there exists a $\rho$-zCDP algorithm that takes an input a dataset $S \subset \mathcal{X}^n$ and w.p.$\geq 1 - \beta$ returns a $((1+\gamma)r, \beta/2k)$-stable point provided that $n \geq k \cdot m(r, \beta/2k)$ for $k = \frac{C\left(\sqrt{d} + \sqrt{\log(\log(1/\beta)/\gamma)}\right)\sqrt{\log(1/\gamma)\log(1/\beta)}}{\gamma\sqrt{\rho}}$. Furthermore, if finding $f(S')$ for any $S'$ containing $m(r, \beta/2k)$-many datapoint takes $\mathsf{T}$ time, then our algorithm runs in time $O(k\mathsf{T} + k \cdot \frac{\log^2(1/\gamma)\log(1/\beta)}{\gamma^2})$.*

*Proof.* The proof simply partitions the $n$ inputs points of $S$ into $k$ disjoint and random subsets $S_1', S_2', ..., S_k'$. W.p. $\geq 1 - \beta/2$ it holds that $\|f(S_i') - c\| \leq r$ for every subset $S_i'$, and then we apply our $(1 + \gamma)$ approximation over this dataset of $k$ many points (with a failure probability of $\beta/2$) and returns the resulting center-point. $\square$

This results improves on Theorem 18 of [21] in both the runtime and the required number of subsamples, at the expense of requiring *all* subsamples to be close to the point $p$ rather than just many of the points.

# D  Missing Proofs: Local-DP Algorithm

**Claim D.1.** Algorithm 5 is a local-model $\rho$-zCDP.

*Proof.* The proof is very similar to the proof of Lemma C.1 — where we apply basically the same accounting, noticing that each $x \in P$ is in charge of randomizing her own data, making this algorithm LDP. $\square$

**Lemma D.2.** W.p. $\geq 1 - \beta$, applying Algorithm 4 with $r \geq r_{opt}$ and an initial center $\theta_0$ s.t. $\|\theta_0 - \theta_{opt}\| \leq 10r_{opt}$ returns a point $\theta^t$ where $|P \setminus B(\theta^t, (1+\gamma)r)| \leq 88\sqrt{\frac{nRT}{\rho}}\left(\sqrt{d} + \sqrt{2\ln(4nRT/\beta_0)}\right) + \sqrt{\frac{2R(T+1)\log(4R(T+1)/\beta_0)}{\rho}}$.

*Proof.* Analogously to the proof of Lemma 4.1, we use the similar definitions: in each iteration $t$ we denote $n_w^t$ as the true number of datapoints in $P$ outside the ball $n_w^t = |\{x \in P : x \notin B(\theta^t, r)\}|$,[8] $\mu_w^t$ as their true mean $\mu_W^t = \frac{1}{n_w^t}\sum_{x \notin B(\theta^t, r)} x$, and $v_w^t$ as the difference of the true mean and the current center $v_w^t = \mu_w^t - \theta^t = \frac{1}{n_w^t}\sum_{x \notin B(\theta^t, r)}(x - \theta^t)$. We thus define the events

$$\mathcal{E}_1 := \text{in all } T+1 \text{ iterations}, |\tilde{n}_w^t - n_w^t| \leq \sqrt{\frac{2nR(T+1)\log(4(T+1)/\beta)}{\rho}}$$

$$\mathcal{E}_2 := \text{in all } T \text{ iterations}, \|\sum_x Z_x^t - n_w^t v_w^t\| \leq \frac{88r\sqrt{nRT}}{\sqrt{\rho}}\left(\sqrt{d} + \sqrt{2\ln(4T/\beta)}\right)$$

Proving that both $\Pr[\overline{\mathcal{E}_1}] \leq \beta/2$ and $\Pr[\overline{\mathcal{E}_2}] \leq \beta/2$ is rather straight-forward. In each iteration $t$ it holds that $\sum_x Y_x^t \sim \mathcal{N}(n_w^t, n\sigma_{count}^2)$ as the sum on $n$ independent Gaussians, and so we merely apply standard Gaussian concentration bounds together with the union bound over all $T+1$ iterations. Similarly, in each iteration $t$ it holds that $\sum_x Z_x^t \sim \mathcal{N}(n_w^t(\mu_x^t - \theta^t), n\sigma_{sum}^2 I_d)$. So standard bounds on the concentration of the $\chi_d^2$-distribution assert that the $L_2$-distance between the random draw from such a $d$-dimensional Gaussian and its mean is $> \sqrt{n\sigma_{sum}^2}(\sqrt{d} + \sqrt{2\ln(4T/\beta)})$ w.p. $< \frac{\beta}{2T}$, after which we apply the union-bound on all $T$ iterations. We continue the rest of the proof conditioning on both $\mathcal{E}_1$ and $\mathcal{E}_2$ holding.

---

[8]Where technically, in the last steps of the algorithm, $n_w^T = |\{x \in P : x \notin B(\theta^T, (1+\gamma)r)\}|$.

Again, due to our if-condition, we make an update-step only when $\tilde{n}_w^t$ is large, which, under $\mathcal{E}_1$ implies that

$$n_w^t \geq \frac{88\sqrt{nRT}}{\sqrt{\rho}}\left(\sqrt{d} + \sqrt{2\ln(4RT/\beta_0)}\right) - \sqrt{\frac{2nR(T+1)\ln(4R(T+1)/\beta_0)}{\rho}} \geq 44|\Delta_{count}|$$

and then proving that the distribution which we use to make an update-step satisfies the conditions detailed in (2) w.h.p. is precisely the same proof (using the independence of $\Delta_{count}$ and $\Delta_{sum}$ and the fact that $\mathbb{E}[\Delta_{sum}] = 0$).

Invoking Corollary 3.5 we have that if we make all $T$ updates then indeed $\|\theta^T - \theta_0\| \leq \gamma r$ and so $|P \setminus B(\theta^T, (1+\gamma)R)| = 0$. So under $\mathcal{E}_1$ Algorithm 4 returns $\theta^T$. Otherwise, at some iteration we do not make an update step, which under $\mathcal{E}_1$ suggest that

$$n_w^t = |P \setminus B(\theta^t, r)| \leq 88\sqrt{\frac{nRT}{\rho}}\left(\sqrt{d} + \sqrt{2\ln(4nRT/\beta_0)}\right) + \sqrt{\frac{2R(T+1)\log(4R(T+1)/\beta_0)}{\rho}}$$

$\square$

**Corollary D.3.** Algorithm 3 altered so it invokes $B = O(\log(1/\gamma))$ calls to Algorithm 5 (instead of Algorithm 4) is a $O(\frac{\log(1/\beta)\log^2(1/\gamma)}{\gamma^2})$-rounds $\rho$-zCDP algorithm in the local-model that takes $O(n \cdot \frac{\log^2(1/\gamma)\log(1/\beta)}{\gamma^2})$-time; and that returns a ball $B(\theta^*, r^*)$ such that $r^* \leq (1+3\gamma)r_{opt}$ and

$$|P \setminus B(\theta^*, r^*)| = O\left(\frac{\sqrt{n}\left(\sqrt{d} + \sqrt{\log(\log(1/\beta)/\gamma)}\right)\sqrt{\log(1/\gamma)\log(1/\beta)}}{\gamma\sqrt{\rho}}\right).$$

*Proof.* The proof follows from using the bound of Lemma D.2 with $T = O(\gamma^{-2}\log(1/\gamma))$, and with a privacy budget of $\rho/B$ and failure probability of $\beta/B$ in each invocation of Algorithm 5. $\square$

# E Experiments

In this section we give an experimental evaluation of our algorithm on three synthetic datasets and one real dataset. We emphasize that our experiment should be perceived merely as a proof-of-concept experiment aimed at the possibility of improving the algorithm's analysis, and not a thorough experimentation for a ready-to-deploy code. We briefly explain the experimental setup below.

**Goal.** We set to investigate the performance of our algorithm, and seeing whether the performance is similar across different types of input and across a range of parameters. In addition, we wondered whether in practice our algorithm halts prior to concluding all $T = O(\gamma^{-2}\ln(1/\gamma))$ iterations.

**Experiment details.** We conducted experiments solely with Algorithm 4 with update-step that uses a constant learning rate of $\gamma^2/8$, feeding it the true $r_{opt}$ of each given dataset as its $r$ parameter. By default, we used the following set of parameters. Our domain in the synthetic experiments is $[-5, 5]^{10}$ (namely, we work in the 10-dimensional space), and our starting point $\theta_0$ is the origin. The default values of our privacy parameter is $\rho = 0.3$, of the approximation constant is $1.2$ (namely $\gamma = 0.2$), and of the failure probability is $\beta = e^{-9} \approx 0.00012$. We set the maximal number of repetitions $T$ just as detailed in Algorithm 4, which depends on $\gamma$.

We varied two of the input parameters, $\rho$ and $\gamma$, and also the data-type. We ran experiments with $\rho \in \{0.1, 0.3, 0.5, 0.7, 0.9\}$ and with $\gamma \in \{0.1, 0.2, 0.3, 0.4, 0.5\}$. Based on the values of $\rho$ and $\gamma$ we computed $n_0 = \frac{\sqrt{RT}(\sqrt{d} + \sqrt{\ln(4RT/\beta_0)})}{\sqrt{\rho}}$ which we used as our halting parameter. In all experiments involving a synthetic dataset, we set the input size $n$ to be $n = 640n_0$.

We varied also the input type, using 3 synthetically generated datasets and one real-life dataset:

- Spherical Gaussian: we generated samples from a $d$-dimensional Gaussian $\mathcal{N}(v, I_d)$, where $v \in \mathbb{R}^d$ is a random shift vector. We discarded each point that did not fall in $[-5, 5]^{10}$.
- Product Distribution: we generated samples from a $d$-dimensional Bernoulli distribution with support $\{-1, 1\}^d$ with various probabilities for each dimension — where for each

coordinate $i \in [10]$ we set $\Pr[x_i = 1] = 2^{-i}$. This creates a "skewed" distribution whose mean is quite far from its 1-center. In order for the 1-center not to coincide with $\theta_0 = \bar{0}$ we shifted this cube randomly in the grid.

- Conditional Gaussian: we repeated the experiment with the spherical Gaussian only this time we conditioned our random draws so that no coordinate lies in the $[0, 0.5]$-interval. This skews the mean of the distribution to be $< 0$ in each coordinate, but leaves the 1-center unaltered. Again, we shifted the Gaussian to a random point $v \in [-5, 5]^d$.

- "Bar Crawl: Detecting Heavy Drinking": a dataset taken from the freely available UCI Machine Learning Repository [1] which collected accelerometer data from participants in a college bar crawl [26]. We truncated the data to only its 3 $x$-, $y$- and $z$-coordinates, and dropped any entry outside of $[-1, 1]^3$, and since it has two points $(-1, -1, -1)$ and $(1, 1, 1)$ then its 1-center is the origin (so we shifted the data randomly in the $[-5, 5]^3$ cube). This left us with $n = 12,921,593$ points. Note that the data is taken from a very few participants, so our algorithm gives an event-level privacy [17].

We ran our experiments in Python, on a (fairly standard) Intel Core i7 2.80 GHz with 16GB RAM and they run in time that ranged from 15 seconds (for $\gamma = 0.5$) to 2 hours (for $\gamma = 0.1$).

**Results.** The results are given in Figures 2, 3, where we plotted the distance of $\theta^t$ to $\theta_{opt}$ for each set of parameters across $t = 10$ repetitions. As evident, we converged to a good approximation of the MEB in all settings. We halt the experiment (i) if $\|\theta_t - \theta_{opt}\| \leq \gamma r_{opt}$, or (ii) if there are not enough wrong points, or (iii) if $t > 2500$ indicating that the run isn't converging. Indeed, the number of iterations until convergence does increase as $\gamma$ decreases; but, rather surprisingly, varying $\rho$ has a small effect on the halting time. This is somewhat expected as $T$ has no dependency on $\rho$ whereas its dependency on $\gamma$ is proportional to $\gamma^{-2}$, but it is evident that as $\rho$ increases our mean-estimation in each iteration becomes more accurate, so one would expect a faster convergence. Also unexpectedly, our results show that even for datasets whose mean and 1-center aren't close to one another (such as the Conditional Gaussian or Product-Distribution), the number of iterations until convergence remains roughly the same (see for example Figure 2 vs. 3).

**Conclusions.** Our experiments suggest that indeed our bound $T$ is a worst-case bound, where in all experiments we concluded in about $7 - 50$ times faster than the bound of Algorithm 4. This suggests that perhaps one would be better off if instead of partitioning the privacy budget equally across all $T$ iterations, they devise some sort of adaptive privacy budgeting. (E.g., using $3\rho/4$ budget on the first $T/4$ iterations and then the remaining $\rho/4$ budget on the latter $3T/4$ iterations.) Such adaptive budgeting is simple when using zCDP, as it does not require "privacy odometers" [33].

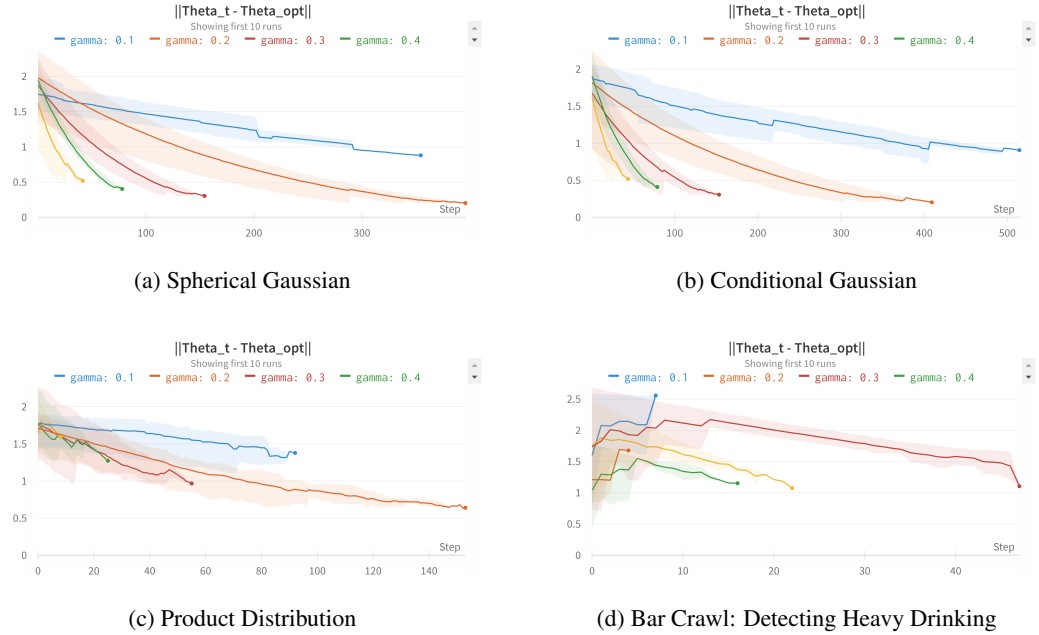

Figure 2: The distance of $\theta^t$ to $\theta_{opt}$ as a function of $t$ – the iteration number, for $\rho = 0.3$ and $\gamma \in \{0.1, 0.2, 0.3, 0.4, 0.5\}$. Each curve corresponds to a different $\gamma$ value. In all experiments the number of iterations until convergence does increase as $\gamma$ decreases, except for $\gamma = 0.1$ where it halts because there were not enough wrong points. Note that for $\gamma = 0.1$ for Bar Crawl dataset (figure 2d) we didn't converge due to its size.

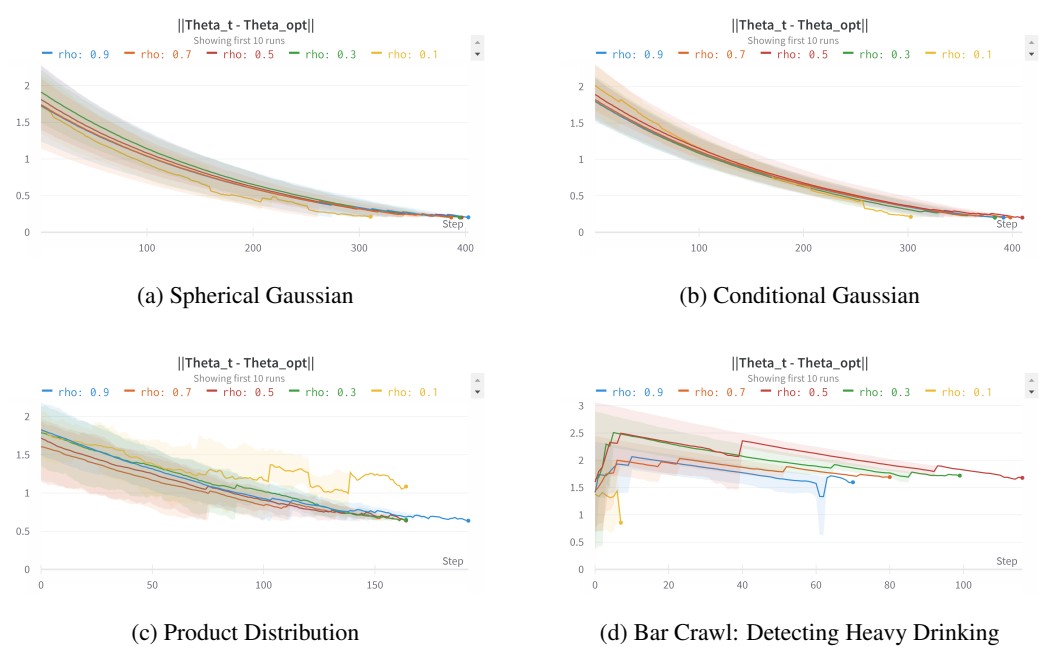

Figure 3: The distance of $\theta^t$ to $\theta_{opt}$ as a function of $t$ – the iteration number, for $\gamma = 0.2$ and $\rho \in \{0.1, 0.3, 0.5, 0.7, 0.9\}$. Each curve corresponds to a different $\rho$ value. In all experiments varying $\rho$ has a small effect on the halting time.