# OpenReview forum: "A Differentially Private Linear-Time fPTAS for the Minimum Enclosing Ball Problem"
_NeurIPS.cc/2022/Conference — NeurIPS 2022 Accept_

### Official Review · Reviewer_kQJZ · 2022-06-14

**Rating:** 7
**Confidence:** 4
**Soundness:** 3 good
**Presentation:** 2 fair
**Contribution:** 3 good

**Summary:**

This paper studies the minimum enclosing ball (MEB) problem under the differential privacy (DP) constraints. In MEB, we are given $n$ points in $\mathbb{R}^d$ and the goal is to output a ball $B(\theta, r)$ of smallest possible radius $r$ that contains all points. In the non-private setting, it is known that this problem is NP-hard and therefore approximation algorithms have been studied. On this front, it is known that there is a fully-polynomial time approximation scheme (FPTAS) for approximating $r$; in other words, there is a $(n/\gamma)^{O(1)}$-time algorithm that outputs a ball of radius at most $(1 + \gamma) * r_{OPT}$ that covers all points.

Differential privacy enforces that the distribution of the output of the algorithm remains "similar" when a single input point is changed. This similarity is paramterized by $\epsilon, \delta \geq 0$. The smaller they are the more privacy protection the algorithm offers. Under DP, it is impossible to hope for a ball that covers all points, and we need to allow a certain number of points to be left out as well. Prior to this paper, the best known $(\epsilon, \delta)$-DP algorithm for MEB runs in time $n^{O(1/\gamma^2)}$ and leaves out $\tilde{O}(\sqrt{d}/\epsilon \gamma^3)$ points (Ghazi et al., NeurIPS 2020). The main result of this paper is an improvement to this, where the running time is now $(n/\gamma)^{O(1)}$ (i.e. the algorithm is an FPTAS) and the number of points left out is only $\tilde{O}(\sqrt{d}/\epsilon \gamma^2)$.

## Non-Private Algorithm

The non-private version of the algorithm is as follows. First, note that we can easily binary search the radius at the cost of $O(\log(1/\gamma))$, so henceforth we will assume that we know $r \approx r_{OPT}$. The algorithm starts with some initial guess for the ball $B(\theta_0, r)$. At iteration $i$, the algorithm produces a new center $\theta_i$ by taking a weighted average between $\theta_{i - 1}$ and the average of the point that lie outside of $B(\theta_{i - 1}, (1 + \gamma) r)$. (This is a "perceptron-like" update.) The authors show that such an algorithm stops in $O(1/\gamma^2)$ steps (meaning that there is no point out side $B(\theta_i, (1 + \gamma) r)$; the analysis is quite similar to the perceptron algorithm.

## Central DP Algorithm

Note that the above algorithm is not yet private. To make it private, the authors add appropriately calibrated Gaussian noise to the sum of points and the number of points outside of the ball $B(\theta_{i - 1}, (1 + \gamma) r)$. This makes the average private. In the actual analysis, Concentrated Differential Privacy (CDP) is used for composition across iteration.

## Local DP Algorithm

Finally, the authors also observe that the above algorithm can be made to work in the local differential privacy (LDP) setting, in which there is no trusted central server and each user has to send messages that are DP. In this setting, it is still possible to compute the sum of points and the number of points privately, by letting each user adds noise. This increases the error by a factor of $O(\sqrt{n})$ and thus in this setting the algorithm may leave out as many as $\tilde{O}(\sqrt{nd}/\epsilon \gamma^2)$ points. Nonetheless, this is already an improvement on the previous bound of $O(n^{0.67})$ (Nissim-Stemmer, ALT 2018) when $d$ is not too large. Furthermore, Nissim-Stemmer work only achieves some (large) constant factor approximation whereas the current work has an approximation ratio arbitrarily close to one.

**Questions:**

## Questions

- Although it is fine to consider approximate-DP, in some cases, pure-DP are still preferred. Can you please discuss whether you algorithms also work in the pure-DP setting and what the bounds are? (My guess is that there is an increase of $\sqrt{d}$ factor in the points left out...)

- It is mentioned (Line 42) that minimizing hinge-loss would not work because the utility gets worse with $1/r$. I'm not sure however whether I agree with this: if you already find an initial ball of size $r$, then isn't it true that you can always just project all the points into such a ball and rescale so that this ball has radius one? Now, you have the problem instance on $r = 1$, right? Can you please comment on why this does not work?

## Other Comments for Authors

- It might be worthwhile mentioning that the LDP algorithm is multi-round, with the number of rounds growing as the number of iterations i.e. $O(1/\gamma^2)$. On the other hand, the algorithm of Nissim-Stemmer works in a fixed number of rounds.

- Another pointer is the work "Efficient Learning of Linear Perceptrons" of Ben-David and Simon (NeurIPS 2000). This paper gives a reduction between the MEB and linear classification with margin. The latter of course can be solved by perceptron. It might be possible to indirectly derive your algorithm from their reduction?

**Limitations:**

No ethical concerns about this work as far as I can tell.

**Strengths And Weaknesses:**

## Strengths

- The algorithm is simple and thus has more chance of being practical.
- The paper is also written clearly and is easy to follow.
- The bounds proved in the paper improve upon several recent works on the topic.

## Weaknesses

-  I think the paper should discuss the known non-private algorithms more clearly. For example, the simplest non-private algorithm is almost the same as the current algorithm, except that the update is not using all points outside of the ball, but rather just the furthest point. In that case, the algorithm converges in $O(1/\epsilon)$ steps; it is unclear what is the relationship between this and the current result in the paper. Moreover, "perceptron-like" algorithms for MEB are also known, see e.g. (Clarkson-Hazan-Woodruff, FOCS'10). It is unclear how similar / different these algorithms are from the one given in the paper.
- Another point is that, all previous works actually consider the more challenging 1-cluster problem, where the goal is only to find a ball that contains at least $t$ points (where $t$ is given as part of the input). MEB is a special case of 1-cluster when $n = t$. Therefore, the results are not directly comparable.

## Overall Evaluation

Overall, I think the algorithm presented in this paper together with its analysis is a nice addition to the tools on clustering / data analytics in DP, even though the writing (especially discussion of previous works) can be improved. Nonetheless, I think the paper can be accepted to NeurIPS.

---

> ### Author Response · Authors · 2022-08-02
> **Authors' Response**
>
> First, we wish to address all reviewers:
> We thank you all for your time and effort in the review and we truly appreciate all of your comments.
> * A recurring comment was regarding the transition from the MEB problem to k-center/means: As we comment ourselves in lines 104-7, we do not know of a way to extend this work to the densest ball problem without first finding a good candidate ball / voronoi region and restricting ourselves to dealing solely with the points within it. In that essence, comparing to previous works is like "comparing apples to oranges."
> * Another comment was about the presence of a similar analysis before, seeing the simplicity of our analysis. We too suspect that a similar analysis was conducted before somewhere, but we did try to find one and could not. We conjecture that such an analysis wasn't published because this SQ-algorithm for the MEB problem is worse on any aspect in comparison to existing fPTAS for the MEB ([Goel et al 00, Badoiu and Clarkson 03]) that make only 1/\gamma^2 steps (a poly-log factor smaller than ours) and are faster. The only advantage (we can see) this SQ-algorithm has over existing baselines is its robustness to the added noise of DP.
> * Indeed, some correctly commented that our LDP algorithm is adaptive and makes O(1/\gamma^2) rounds. That is correct and we will add it to the paper's final version, as well as an open problem to get a O(1)-rounds algorithm (independent of \gamma).
>
>
>  Second, to answer your individual comments:
> * First, we believe that was a technical typo in your review, as the best non-private algorithms (we know of) from the MEB problem converge in 1/\epsilon^2 iterations (not 1/\epsilon).
> * Many thanks for the reference to [CHW10]. But it is unlikely that our work mimics their approach as their analysis uses (a sublinear variant of) the MW algo and relies on off-the-shelf regret bounds for convex optimization, whereas ours uses the simple geometric insight from Claim 3.3. In addition, their approximation is an additive one rather than a multiplicative one.
> * Conversion to Pure-DP: Indeed, the \sqrt d changes to linear in d dependency. Moreover, per iteration our privacy budget is \rho_{iter} = \rho/\gamma^2, and since our overall bounds are dependent on \sqrt{1/ \rho_{iter} } the current bounds yield \gamma^{-2} overall. If we were to convert to pure-DP then our per-iteration privacy-budget is \epislon_{iter}=\epsilon/\gamma^2, and the dependency on 1/\epsilon_{iter} yields an overall dependency on \gamma^{-3}. As it currently stands, we do not think that including a Pure-DP analysis of our algorithm would improve the paper, as \epsilon-DP and \rho-zCDP have very similar composition. However, we welcome further feedback on this --- so if you think we ought to include this pure-DP analysis in the final version of our paper, please let us know.
> * Many thanks as well for the reference to Ben-David and Simon (a work we didn't know before). It is a very interesting work, but our understanding is that it actually shows a reduction in the opposite direction (from hyperplane to Densest Ball and then uses an off-the-shelf algorithm for the Densest Ball problem). It is due to your reference that we wish to give more serious and thorough investigation as to the possible equivalence between the two problems and its applicability under DP.
> * Your proposed ERM algorithm is another spot-on comment. Indeed, it is possible to interpret our work as saying that the SGD for the hinge-loss of the L_2-difference (i.e. \max\{ 0, \frac  {\|\theta-x\|^2 - r_i^2} {r_{global}^2}\} ) requires only a constant number (\tilde O(1/\gamma^2)) of non-zero-gradient steps, thus yielding our linear time algorithm rather then a quadratic time "vanilla" DP-SGD.  We plan on further discussing this in the final version of our work.
> * On a personal note, your review is one of the most insightful reviews we *ever* received, one that has given us multiple points to think about. It was a treat to read and we truly and wholeheartedly appreciate it. We really tip our hats to you!

---

> > ### Comment · Reviewer_kQJZ · 2022-08-03
> > **Thank you**
> >
> > I'd like to thank the authors for the rebuttal. They resolve my earlier confusions / concerns.
> >
> > Indeed, you're right that the non-private algorithm (i.e. the one that keeps adding the furthest point to the set) needs $O(1/\gamma^2)$ iteration. When I was writing the comment, I was thinking about $O(1/\gamma)$-size *coresets*. But the constructions of such coresets are more complicated (e.g. they use approximation algorithm for MEB as subroutine as so it is unclear how to count the number of iterations) so they are not comparable.

---

### Official Review · Reviewer_nRv1 · 2022-07-10

**Rating:** 6
**Confidence:** 3
**Soundness:** 3 good
**Presentation:** 2 fair
**Contribution:** 3 good

**Summary:**


This paper presents a differentially private algorithm for approximating the ball with the smallest radius that encloses all the input points in an arbitrary dimension. The classical approximation algorithm for this problem runs iteratively. In each iteration, it picks the farthest point from the current center and moves it this direction by a certain “step size”. The step size is smaller for higher iteration and the algorithm runs in $O(n/\gamma)$ time to compute a (1+\gamma)-approximation. Unfortunately, the execution of this algorithm relies on the location of the farthest point and therefore it is non-private.

To overcome this difficulty, this paper modifies the non-private algorithm as follows. Instead of picking the farthest point, this paper takes the mean of all the points that are currently not covered by a ball of optimal radius (such a radius can be obtained by conducting O(\log n) guesses) and then moves towards the center of the ball towards this mean. The authors show that after sufficient iterations, the estimated center comes $\gamma$ close to the optimal center as desired.
Next, given that the mean is sufficient, the authors are able to extend their algorithm to differentially private setting and the local DP setting (although they have to guarantee DP, they have to ignore a small number of points).


**Questions:**


- The original paper for MEB approximation gave extension of their approximation methods to the k-center problem. Can your algorithm also be extended to the k-center problem? If not, why?

- The best-known algorithm for MEB approximation is simply an instance of Frank-Wolfe gradient descent method. The authors result rely heavily on MEB properties and hence, the new DP algorithm does not give a modified FW-algorithm that is differentially private. Is this correct?


**Strengths And Weaknesses:**

strengths:
- There is a substantial improvement in the execution time over the previous DP algorithm for the MEB problem. MEB problem is closely linked to several important problems in ML, for instance, SVM.
- Although slower than the previous non-private algorithms, the non-private approximation algorithm presented here appears new and nice.

weakness:
- the authors have not presented an implementation of their algorithm. Perhaps, the constants make the algorithm impractical?
- the authors have not considered extending these algorithms to the k-center clustering problem or the SVM problem both of which are of value to the ML community.
- Presentation of the results can be better. For instance, instead of stating the very minor differences between algorithms 1 and 3, the authors basically repeat the whole algorithm which takes up a lot of space.

---

> ### Author Response · Authors · 2022-08-02
> **Authors' Response**
>
> First, we wish to address all reviewers:
> We thank you all for your time and effort in the review and we truly appreciate all of your comments.
> * A recurring comment was regarding the transition from the MEB problem to k-center/means: As we comment ourselves in lines 104-7, we do not know of a way to extend this work to the densest ball problem without first finding a good candidate ball / voronoi region and restricting ourselves to dealing solely with the points within it. In that essence, comparing to previous works is like "comparing apples to oranges."
> * Another comment was about the presence of a similar analysis before, seeing the simplicity of our analysis. We too suspect that a similar analysis was conducted before somewhere, but we did try to find one and could not. We conjecture that such an analysis wasn't published because this SQ-algorithm for the MEB problem is worse on any aspect in comparison to existing fPTAS for the MEB ([Goel et al 00, Badoiu and Clarkson 03]) that make only 1/\gamma^2 steps (a poly-log factor smaller than ours) and are faster. The only advantage (we can see) this SQ-algorithm has over existing baselines is its robustness to the added noise of DP.
> * Indeed, some correctly commented that our LDP algorithm is adaptive and makes O(1/\gamma^2) rounds. That is correct and we will add it to the paper's final version, as well as an open problem to get a O(1)-rounds algorithm (independent of \gamma).
>
>
>
> Second, to answer your individual comments:
> * The algorithm was implemented and we conducted a few rudimentary experiments with the algorithm, as detailed in the Supplementary Material, Section D. In fact, the goal of our experiments was, in part, to try and assess how far off are our worst-case bounds (and constants) from the bounds required for successful implementation.
> * Presentation: While this point mainly comes down to individual view, we believe that it is rather evident we dedicated serious thought to the problem's presentation, starting with the non-private non-noisy setting and "working our way up" towards a DP algorithm. Some reviewers appreciated it, some didn't. We understand that not everybody would agree that this is the best way to present the work, but it is the presentation we decided upon after serious deliberation. But that is not to say that our presentation cannot be improved --- in fact, we'd welcome concrete suggestions as to how to improve on the paper's presentation and we promise to give them serious consideration.
> * Do we get a "new" FW-SGD algorithm? Indeed, you are right and we do not know of a general way to convert this MEB problem to a general convex problem. Mostly because we use the simple geometric insight of Claim 3.3 for which we do not know an analogue in the general case.

---

> > ### Comment · Reviewer_nRv1 · 2022-08-08
> > **Follow-up**
> >
> > Thank you for answering my questions. Based on your responses, I am happy to raise my score.
> >
> > I would recommend reading (also adding) the reference [1]. Paper [1] shows how the coreset construction for MEB and many other problems are simply special instances of Frank-Wolfe algorithm and using a simple farthest point heuristic leads to 1/\gamma sized coreset for MEB. Perhaps, ideas from your paper can be combined with the analysis here to generalize your result to many other problems.
> >
> > Regarding presentation: I am very well versed with MEB literature but have very limited exposure to differential privacy. As a reader with such a background, I would have loved to see more explanation (in the main text) on why these algorithms are differentially private.
> >
> >
> > [1] Kenneth L. Clarkson: Coresets, sparse greedy approximation, and the Frank-Wolfe algorithm. ACM Trans. Algorithms 6(4): 63:1-63:30 (2010)

---

### Official Review · Reviewer_qMEV · 2022-07-10

**Rating:** 7
**Confidence:** 4
**Soundness:** 4 excellent
**Presentation:** 3 good
**Contribution:** 3 good

**Summary:**

This paper studies the Minimum Enclosing Ball problem (aka 1-Center) in the Differential Privacy Setting. In this problem, you have a set of data points X, and you need to find a point p such that if all the points X fit in some ball of radius r, then the ball of radius (1+gamma)*r around p contains X. This paper provides the first private FPTAS for this problem, meaning that it (approximately) succeeds in the task with high probability, running in time roughly O(nd/gamma^2) time. The paper includes an algorithm that works in the central model of privacy, and another one in the local model of privacy. As in most privacy papers, their accuracy guarantee is not exact: the ball of radius (1+gamma)*r they find may omit up to roughly sqrt{d} points in the central privacy setting, and up to sqrt{n*d} points in the local privacy setting. Previous work of Ghazi-Kumar-Manurangsi provided a PTAS for this problem, meaning the runtime was poly(n, d) when gamma > 0 is a fixed constant. The previous work was applied as a subroutine for private clustering.

Finally, there also appear to be some experiments: I didn't read this part as they are in the appendix and I was mainly interested in the theoretical aspect of this paper, but this is worth mentioning in case other reviewers are interested in them and didn't notice them.


**Questions:**

1) Is your algorithm for the non-private MEB different from the previous FPTAS algorithms? The algorithm is in fact remarkably simple so I am quite surprised this part is new. If it is in fact based on previous work you should mention this at the beginning of Section 3.
2) Am I correct in saying that your LDP algorithm requires several rounds of interactivity? If so, you should explicitly mention this.

**Limitations:**

N/A (This work is almost entirely theoretical).

**Strengths And Weaknesses:**

1) The question of 1-center is very interesting in my opinion as it is one of the simplest problems in computational geometry. This problem also has lots of applications (as the authors mention) to problems in clustering, tuning SVM parameters, etc. So, obtaining a FPTAS for this problem with differential privacy is a very nice result.
2) The technique is extremely simple (especially in the non-private setting), which should make it very easy and beneficial to implement in practice. The private algorithm is of course more complicated than the non-private one, but overall is surprisingly simple.
3) One downside of their method is that it doesn't work in the case of significant outliers (such as in the setting that Ghazi-Kumar-Manurangsi use to apply to private k-means). Indeed, the authors acknowledge this as a shortcoming, and obtaining an FPTAS for a k-means coreset appears to be a very interesting open problem.

---

> ### Author Response · Authors · 2022-08-02
> **Authors' Response**
>
> First, we wish to address all reviewers:
> We thank you all for your time and effort in the review and we truly appreciate all of your comments.
> * A recurring comment was regarding the transition from the MEB problem to k-center/means: As we comment ourselves in lines 104-7, we do not know of a way to extend this work to the densest ball problem without first finding a good candidate ball / voronoi region and restricting ourselves to dealing solely with the points within it. In that essence, comparing to previous works is like "comparing apples to oranges."
> * Another comment was about the presence of a similar analysis before, seeing the simplicity of our analysis. We too suspect that a similar analysis was conducted before somewhere, but we did try to find one and could not. We conjecture that such an analysis wasn't published because this SQ-algorithm for the MEB problem is worse on any aspect in comparison to existing fPTAS for the MEB ([Goel et al 00, Badoiu and Clarkson 03]) that make only 1/\gamma^2 steps (a poly-log factor smaller than ours) and are faster. The only advantage (we can see) this SQ-algorithm has over existing baselines is its robustness to the added noise of DP.
> * Indeed, some correctly commented that our LDP algorithm is adaptive and makes O(1/\gamma^2) rounds. That is correct and we will add it to the paper's final version, as well as an open problem to get a O(1)-rounds algorithm (independent of \gamma).
>
> We believe that these address the concerns you have raised. If there are others (and NeurIPS makes it possible) then we'd be happy to answer those too.

---

### Official Review · Reviewer_VWyJ · 2022-07-11

**Rating:** 7
**Confidence:** 2
**Soundness:** 3 good
**Presentation:** 4 excellent
**Contribution:** 3 good

**Summary:**

This work studies the Minimum Enclosing Ball problem, where we are given a set P of n points in d-dimensional Euclidean space and the objective is to output a ball that contains all these points, such that the radius is minimized. This problem has various applications and is also related to the notion of core-sets in the clustering literature. Prior works have proposed fully polynomial time approximation schemes (fPTAS) with various guarantees. This work considers differentially private algorithms (DP) and obtains an DP-fPTAS, which seems to be the first DP-fPTAS guarantee for this problem. More precisely, the algorithm outputs a ball whose radius is a (1 + gamma) approximation in (n / \gamma^2) time (ignoring log factors) while leaving out at most (sqrt(d) / (\gamma^2 \epsilon)) points (again ignoring log factors) where \epsilon depends on the privacy-parameter. The idea is fairly simple: Start with a good initialization for the ball (using ideas from prior works) and iteratively improve it using a perceptron-style algorithm. This algorithm is also robust to noisy mean-estimation subroutines, therefore this can be exploited to give DP algorithms. This is done for both curator-model as well as local-model differential privacy. As the authors point out, the latter guarantee improves on prior works as far as the dependence on n goes but has worse dependence on other parameters. Finally, simple experiments are conducted to validate the theory. Experiments include synthetic datasets (Gaussian and product of biased Bernoulli distributions) and a dataset containing accelerometer data from a college bar crawl. The experiments plot the distance of the predicted ball center from the optimal center as a function of number of iterations and are meant to be a proof-of-sketch of the theory.

**Questions:**

Apart from the comments above, other suggestions are as follows

1. In the first paragraph, it would be good to clarify what norm is being used for the balls

2. In L33-36, what is the role of \epsilon? How does it relate to the privacy parameter?

3. In L38, "a" should be "as"

4. In the third line of algorithm 4, "raduis" should be "radius"

5. L380: "have" should be "has"

6. L512: "teach" -> "each"

**Limitations:**

Limitations of the theoretical results are discussed (parameter dependence, assumptions, etc.), but negative societal aspects are not discussed.

**Strengths And Weaknesses:**

I should add a disclaimer that although I work in theoretical computer science, differentially private algorithms are not my field of expertise. However, the theory and proofs seem correct as far as I can tell and the contributions are worth publishing.

### Strengths:

1. The paper is very well-written, the algorithms and proofs are easy to follow. Related work is well-cited, making it clear where the contributions lie.

2. The algorithm is quite easy to describe. In fact, the authors do a great job sketching it to a decent extent in the introduction (L53-107).

### Weaknesses:

1. Code is not provided in the supplementary material, despite the checklist saying it's provided.

2. The experiments are very simple and while they add support to the theory, they are not convincing enough to argue that this is a practical algorithm. However, as the authors mention, this is a theory paper and the experiments are just a proof-of-sketch.

---

> ### Author Response · Authors · 2022-08-02
> **Authors' Response**
>
> First, we wish to address all reviewers:
> We thank you all for your time and effort in the review and we truly appreciate all of your comments.
> * A recurring comment was regarding the transition from the MEB problem to k-center/means: As we comment ourselves in lines 104-7, we do not know of a way to extend this work to the densest ball problem without first finding a good candidate ball / voronoi region and restricting ourselves to dealing solely with the points within it. In that essence, comparing to previous works is like "comparing apples to oranges."
> * Another comment was about the presence of a similar analysis before, seeing the simplicity of our analysis. We too suspect that a similar analysis was conducted before somewhere, but we did try to find one and could not. We conjecture that such an analysis wasn't published because this SQ-algorithm for the MEB problem is worse on any aspect in comparison to existing fPTAS for the MEB ([Goel et al 00, Badoiu and Clarkson 03]) that make only 1/\gamma^2 steps (a poly-log factor smaller than ours) and are faster. The only advantage (we can see) this SQ-algorithm has over existing baselines is its robustness to the added noise of DP.
> * Indeed, some correctly commented that our LDP algorithm is adaptive and makes O(1/\gamma^2) rounds. That is correct and we will add it to the paper's final version, as well as an open problem to get a O(1)-rounds algorithm (independent of \gamma).
>
>
> Second, to answer your individual comments:
> * The missing code: We wholeheartedly apologize and we are deeply embarrassed about it. The code was supposed to be added and we made an honest mistake using an empty "Code" folder. The code is written in Python, it is very simple and we promise that we will make it public.
> * Thank you for pointing out our typos. We will fix them in the paper's final version.

---

### Meta-Review · Area_Chair_eZv7 · 2022-08-24

**Recommendation:** Accept
**Confidence:** Certain

**Metareview:**

The paper improves the state-of-the-art for the minimum enclosing ball problem under differential privacy constraints. Moreover, the algorithm and its analysis are simple and intuitive.

The reviewers agreed that the paper is a concrete advance in the area, and the ideas may lead to a practical implementation. The authors carefully responded to all the issues raised by the reviewers, clearing the way to acceptance.

**Award:**

No

---

### Decision · Program_Chairs · 2022-09-14

Accept